# FAIR RESOURCE ALLOCATION IN FEDERATED LEARNING

**Tian Li**
CMU
tianli@cmu.edu

**Maziar Sanjabi**
Facebook AI
maziars@fb.com

**Ahmad Beirami**
Facebook AI
beirami@fb.com

**Virginia Smith**
CMU
smithv@cmu.edu

## ABSTRACT

Federated learning involves training statistical models in massive, heterogeneous networks. Naively minimizing an aggregate loss function in such a network may disproportionately advantage or disadvantage some of the devices. In this work, we propose $q$-Fair Federated Learning ($q$-FFL), a novel optimization objective inspired by fair resource allocation in wireless networks that encourages a more fair (specifically, a more *uniform*) accuracy distribution across devices in federated networks. To solve $q$-FFL, we devise a communication-efficient method, $q$-FedAvg, that is suited to federated networks. We validate both the effectiveness of $q$-FFL and the efficiency of $q$-FedAvg on a suite of federated datasets with both convex and non-convex models, and show that $q$-FFL (along with $q$-FedAvg) outperforms existing baselines in terms of the resulting fairness, flexibility, and efficiency.

## 1 INTRODUCTION

Federated learning is an attractive paradigm for fitting a model to data generated by, and residing on, a network of remote devices (McMahan et al., 2017). Unfortunately, naively minimizing an aggregate loss in a large network may disproportionately advantage or disadvantage the model performance on some of the devices. For example, although the accuracy may be high on average, there is no accuracy guarantee for individual devices in the network. This is exacerbated by the fact that the data are often heterogeneous in federated networks both in terms of size and distribution, and model performance can thus vary widely. In this work, we therefore ask: Can we devise an efficient federated optimization method to encourage a *more fair (i.e., more uniform) distribution* of the model performance across devices in federated networks?

There has been tremendous recent interest in developing fair methods for machine learning (see, e.g., Cotter et al., 2019; Dwork et al., 2012). However, current approaches do not adequately address concerns in the federated setting. For example, a common definition in the fairness literature is to enforce *accuracy parity* between protected groups[1] (Zafar et al., 2017a). For devices in massive federated networks, however, it does not make sense for the accuracy to be *identical* on each device given the significant variability of data in the network. Recent work has taken a step towards addressing this by introducing *good-intent fairness*, in which the goal is instead to ensure that the training procedure does not overfit a model to any one device at the expense of another (Mohri et al., 2019). However, the proposed objective is rigid in the sense that it only maximizes the performance of the worst performing device/group, and has only be tested in small networks (for 2-3 devices). In realistic federated learning applications, it is natural to instead seek methods that can flexibly trade off between overall performance and fairness in the network, and can be implemented at scale across hundreds to millions of devices.

In this work, we propose $q$-FFL, a novel optimization objective that addresses fairness issues in federated learning. Inspired by work in fair resource allocation for wireless networks, $q$-FFL minimizes an aggregate *reweighted* loss parameterized by $q$ such that the devices with higher loss are given higher relative weight. We show that this objective encourages a device-level definition

---

[1]While fairness is typically concerned with performance between "groups", we define fairness in the federated setting at a more granular scale in terms of the devices in the network. We note that devices may naturally combine to form groups, and thus use these terms interchangeably in the context of prior work.

of fairness in the federated setting, which generalizes standard accuracy parity by measuring the degree of uniformity in performance across devices. As a motivating example, we examine the test accuracy distribution of a model trained via a baseline approach (`FedAvg`) vs. $q$-`FFL` in Figure 1. Due to the variation in the data across devices, the model accuracy is quite poor on some devices. By using $q$-`FFL`, we can maintain the same overall average accuracy while ensuring a more fair/uniform quality of service across the network. Adaptively minimizing our $q$-`FFL` objective results in a flexible framework that can be tuned depending on the desired amount of fairness.

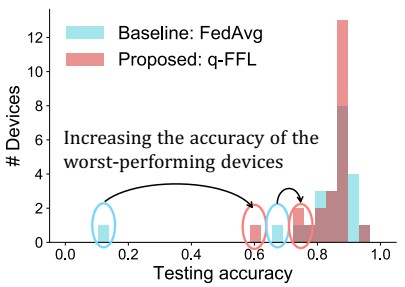

Figure 1: Model performance (e.g., test accuracy) in federated networks can vary widely across devices. Our objective, $q$-`FFL`, aims to increase the fairness/uniformity of model performance while maintaining average performance.

To solve $q$-`FFL` in massive federated networks, we additionally propose a lightweight and scalable distributed method, $q$-`FedAvg`. Our method carefully accounts for important characteristics of the federated setting such as communication-efficiency and low participation of devices (Bonawitz et al., 2019; McMahan et al., 2017). The method also reduces the overhead of tuning the hyperparameter $q$ in $q$-`FFL` by dynamically estimating the step-sizes associated with different values of $q$.

Through extensive experiments on federated datasets with both convex and non-convex models, we demonstrate the fairness and flexibility of $q$-`FFL` and the efficiency of $q$-`FedAvg` compared with existing baselines. In terms of fairness, $q$-`FFL` is able to reduce the variance of accuracies across devices by 45% on average while maintaining the same overall average accuracy. In terms of efficiency, our distributed method, $q$-`FedAvg`, is capable of solving the proposed objective orders-of-magnitude more quickly than other baselines. Finally, while we consider our approaches primarily in the context of federated learning, we also demonstrate that $q$-`FFL` can be applied to other related problems such as meta-learning, helping to produce fair initializations across multiple tasks.

## 2 RELATED WORK

**Fairness in Resource Allocation.** Fair resource allocation has been extensively studied in fields such as network management (Ee & Bajcsy, 2004; Hahne, 1991; Kelly et al., 1998; Neely et al., 2008) and wireless communications (Eryilmaz & Srikant, 2006; Nandagopal et al., 2000; Sanjabi et al., 2014; Shi et al., 2014). In these contexts, the problem is defined as allocating a scarce shared resource, e.g., communication time or power, among many users. In these cases, directly maximizing utilities such as total throughput may lead to unfair allocations where some users receive poor service. As a service provider, it is important to improve the quality of service for all users while maintaining overall throughput. For this reason, several popular fairness measurements have been proposed to balance between fairness and total throughput, including Jain's index (Jain et al., 1984), entropy (Rényi et al., 1961), max-min/min-max fairness (Radunovic & Le Boudec, 2007), and proportional fairness (Kelly, 1997). A unified framework is captured through $\alpha$-fairness (Lan et al., 2010; Mo & Walrand, 2000), in which the network manager can tune the emphasis on fairness by changing a single parameter, $\alpha$.

To draw an analogy between federated learning and the problem of resource allocation, one can think of the global model as a resource that is meant to serve the users (or devices). In this sense, it is natural to ask similar questions about the fairness of the service that users receive and use similar tools to promote fairness. Despite this, we are unaware of any works that use $\alpha$-fairness from resource allocation to modify objectives in machine learning. Inspired by the $\alpha$-fairness metric, we propose a similarly modified objective, $q$-Fair Federated Learning ($q$-`FFL`), to encourage a more fair accuracy distribution across devices in the context of federated training. Similar to the $\alpha$-fairness metric, our $q$-`FFL` objective is flexible enough to enable trade-offs between fairness and other traditional metrics such as accuracy by changing the parameter $q$. In Section 4, we show empirically that the use of $q$-`FFL` as an objective in federated learning enables a more uniform accuracy distribution across devices—significantly reducing variance while maintaining the average accuracy.

**Fairness in Machine Learning.** Fairness is a broad topic that has received much attention in the machine learning community, though the goals often differ from that described in this work. Indeed,

*fairness* in machine learning is typically defined as the protection of some specific attribute(s). Two common approaches are to preprocess the data to remove information about the protected attribute, or to post-process the model by adjusting the prediction threshold after classifiers are trained (Feldman, 2015; Hardt et al., 2016; Calmon et al., 2017). Another set of works optimize an objective subject to some fairness constraints during training time (Agarwal et al., 2018; Cotter et al., 2019; Hashimoto et al., 2018; Woodworth et al., 2017; Baharlouei et al., 2020; Zafar et al., 2017a;b; Dwork et al., 2012). Our work also enforces fairness during training, although we define fairness as the uniformity of the accuracy distribution across devices in federated learning (Section 3), as opposed to the protection of a specific attribute. Although some works define accuracy parity to enforce equal error rates among specific groups as a notion of fairness (Zafar et al., 2017a; Cotter et al., 2019), devices in federated networks may not be partitioned by protected attributes, and our goal is not to optimize for identical accuracy across all devices. Cotter et al. (2019) use a notion of 'minimum accuracy', which is conceptually similar to our goal. However, it requires one optimization constraint for each device, which would result in hundreds to millions of constraints in federated networks.

In federated settings, Mohri et al. (2019) recently proposed a minimax optimization scheme, Agnostic Federated Learning (`AFL`), which optimizes for the performance of the single worst device. This method has only been applied at small scales (for a handful of devices). Compared to `AFL`, our proposed objective is more flexible as it can be tuned based on the desired amount of fairness; `AFL` can in fact be seen as a special case of our objective, $q$-`FFL`, with large enough $q$. In Section 4, we demonstrate that the flexibility of our objective results in more favorable accuracy vs. fairness trade-offs than `AFL`, and that $q$-`FFL` can also be solved at scale more efficiently.

**Federated Optimization.** Federated learning faces challenges such as expensive communication, variability in systems environments in terms of hardware or network connection, and non-identically distributed data across devices (Li et al., 2019). In order to reduce communication and tolerate heterogeneity, optimization methods must be developed to allow for local updating and low participation among devices (McMahan et al., 2017; Smith et al., 2017). We incorporate these key ingredients when designing methods to solve our $q$-`FFL` objective efficiently in the federated setting (Section 3.3).

## 3 FAIR FEDERATED LEARNING

In this section, we first formally define the classical federated learning objective and methods, and introduce our proposed notion of fairness (Section 3.1). We then introduce $q$-`FFL`, a novel objective that encourages a more fair (uniform) accuracy distribution across all devices (Section 3.2). Finally, in Section 3.3, we describe $q$-`FedAvg`, an efficient distributed method to solve the $q$-`FFL` objective in federated settings.

### 3.1 PRELIMINARIES: FEDERATED LEARNING, FEDAVG, AND FAIRNESS

Federated learning algorithms involve hundreds to millions of remote devices learning locally on their device-generated data and communicating with a central server periodically to reach a global consensus. In particular, the goal is typically to solve:

$$\min_w f(w) = \sum_{k=1}^{m} p_k F_k(w), \tag{1}$$

where $m$ is the total number of devices, $p_k \geq 0$, and $\sum_k p_k = 1$. The local objective $F_k$'s can be defined by empirical risks over local data, i.e., $F_k(w) = \frac{1}{n_k} \sum_{j_k=1}^{n_k} l_{j_k}(w)$, where $n_k$ is the number of samples available locally. We can set $p_k$ to be $\frac{n_k}{n}$, where $n = \sum_k n_k$ is the total number of samples to fit a traditional empirical risk minimization-type objective over the entire dataset.

Most prior work solves (1) by sampling a subset of devices with probabilities $p_k$ at each round, and then running an optimizer such as stochastic gradient descent (SGD) for a variable number of iterations locally on each device. These *local updating methods* enable flexible and efficient communication compared to traditional mini-batch methods, which would simply calculate a subset of the gradients (Stich, 2019; Wang & Joshi, 2018; Woodworth et al., 2018; Yu et al., 2019). `FedAvg` (McMahan et al., 2017), summarized in Algorithm 3 in Appendix C.1, is one of the leading methods to solve (1) in non-convex settings. The method runs simply by having each selected device apply $E$ epochs of SGD locally and then averaging the resulting local models.

Unfortunately, solving problem (1) in this manner can implicitly introduce highly variable performance between different devices. For instance, the learned model may be biased towards devices with larger numbers of data points, or (if weighting devices equally), to commonly occurring devices. More formally, we define our desired fairness criteria for federated learning below.

**Definition 1** (*Fairness of performance distribution*). For trained models $w$ and $\tilde{w}$, we informally say that model $w$ provides a more *fair* solution to the federated learning objective (1) than model $\tilde{w}$ if the performance of model $w$ on the $m$ devices, $\{a_1, \ldots a_m\}$, is more *uniform* than the performance of model $\tilde{w}$ on the $m$ devices.

In this work, we take 'performance', $a_k$, to be the *testing accuracy* of applying the trained model $w$ on the test data for device $k$. There are many ways to mathematically evaluate the *uniformity* of the performance. In this work, we mainly use the *variance* of the performance distribution as a measure of uniformity. However, we also explore other uniformity metrics, both empirically and theoretically, in Appendix A.1. We note that a tension exists between the *fairness/uniformity* of the final testing accuracy and the *average* testing accuracy across devices. In general, our goal is to impose more *fairness/uniformity* while maintaining the same (or similar) average accuracy.

**Remark 2** (Connections to other fairness definitions). *Definition 1 targets device-level fairness, which has finer granularity than the classical attribute-level fairness such as accuracy parity (Zafar et al., 2017a). We note that in certain cases where devices can be naturally clustered into groups with specific attributes, our definition can be seen as a relaxed version of accuracy parity, in that we optimize for similar but not necessarily identical performance across devices.*

## 3.2 THE OBJECTIVE: $q$-FAIR FEDERATED LEARNING ($q$-FFL)

A natural idea to achieve fairness as defined in (1) would be to *reweight* the objective—assigning higher weights to devices with poor performance, so that the distribution of accuracies in the network shifts towards more uniformity. Note that this reweighting must be done dynamically, as the performance of the devices depends on the model being trained, which cannot be evaluated a priori. Drawing inspiration from $\alpha$-fairness, a utility function used in fair resource allocation in wireless networks, we propose the following objective. For given local non-negative cost functions $F_k$ and parameter $q > 0$, we define the $q$-Fair Federated Learning ($q$-FFL) objective as:

$$\min_w \ f_q(w) = \sum_{k=1}^{m} \frac{p_k}{q+1} F_k^{q+1}(w) \,, \tag{2}$$

where $F_k^{q+1}(\cdot)$ denotes $F_k(\cdot)$ to the power of $(q+1)$. Here, $q$ is a parameter that tunes the amount of fairness we wish to impose. Setting $q = 0$ does not encourage fairness beyond the classical federated learning objective (1). A larger $q$ means that we emphasize devices with higher local empirical losses, $F_k(w)$, thus imposing more uniformity to the training accuracy distribution and potentially inducing fairness in accordance with Definition 1. Setting $f_q(w)$ with a large enough $q$ reduces to classical minimax fairness (Mohri et al., 2019), as the device with the worst performance (largest loss) will dominate the objective. We note that while the $(q+1)$ term in the denominator in (2) may be absorbed in $p_k$, we include it as it is standard in the $\alpha$-fairness literature and helps to ease notation. For completeness, we provide additional background on $\alpha$-fairness in Appendix B.

As mentioned previously, $q$-FFL generalizes prior work in fair federated learning (AFL) (Mohri et al., 2019), allowing for a flexible trade-off between fairness and accuracy as parameterized by $q$. In our theoretical analysis (Appendix A), we provide generalization bounds of $q$-FFL that generalize the learning bounds of the AFL objective. Moreover, based on our fairness definition (Definition 1), we theoretically explore how $q$-FFL results in more *uniform* accuracy distributions with increasing $q$. Our results suggest that $q$-FFL is able to impose 'uniformity' of the test accuracy distribution in terms of various metrics such as variance and other geometric and information-theoretic measures.

In our experiments (Section 4.2), on both convex and non-convex models, we show that using the $q$-FFL objective, we can obtain fairer/more uniform solutions for federated datasets in terms of both the training and testing accuracy distributions.

### 3.3 THE SOLVER: FEDAVG-STYLE $q$-FAIR FEDERATED LEARNING ($q$-FEDAVG)

In developing a functional approach for fair federated learning, it is critical to consider not only what objective to solve but also how to solve such an objective efficiently in a massive distributed network. In this section, we provide methods to solve $q$-FFL. We start with a simpler method, $q$-FedSGD, to illustrate our main techniques. We then provide a more efficient counterpart, $q$-FedAvg, by considering local updating schemes. Our proposed methods closely mirror traditional distributed optimization methods—mini-batch SGD and federated averaging (FedAvg)—but with step-sizes and subproblems carefully chosen in accordance with the $q$-FFL problem (2).

**Achieving variable levels of fairness: tuning $q$.** In devising a method to solve $q$-FFL (2), we begin by noting that it is crucial to first determine how to set $q$. In practice, $q$ can be tuned based on the desired amount of fairness (with larger $q$ inducing more fairness). As we describe in our experiments (Section 4.2), it is therefore common to train a *family of objectives* for different $q$ values so that a practitioner can explore the trade-off between accuracy and fairness for the application at hand.

One concern with solving such a family of objectives is that it requires step-size tuning for every value of $q$. In particular, in gradient-based methods, the step-size inversely depends on the Lipschitz constant of the function's gradient, which will change as we change $q$. This can quickly cause the search space to explode. To overcome this issue, we propose estimating the local Lipschitz constant for the family of $q$-FFL objectives by using the Lipschitz constant we infer by tuning the step-size (via grid search) on just one $q$ (e.g., $q = 0$). This allows us to dynamically adjust the step-size of our gradient-based optimization method for the $q$-FFL objective, avoiding manual tuning for each $q$. In Lemma 3 below we formalize the relation between the Lipschitz constant, $L$, for $q = 0$ and $q > 0$.

**Lemma 3.** *If the non-negative function $f(\cdot)$ has a Lipschitz gradient with constant L, then for any $q \geq 0$ and at any point $w$,*

$$L_q(w) = Lf(w)^q + qf(w)^{q-1}\|\nabla f(w)\|^2 \tag{3}$$

*is an upper-bound for the local Lipschitz constant of the gradient of $\frac{1}{q+1}f^{q+1}(\cdot)$ at point $w$.*

*Proof.* At any point $w$, we can compute the Hessian $\nabla^2\left(\frac{1}{q+1}f^{q+1}(w)\right)$ as:

$$\nabla^2\left(\frac{1}{q+1}f^{q+1}(w)\right) = qf^{q-1}(w)\underbrace{\nabla f(w)\nabla^T f(w)}_{\preceq\|\nabla f(w)\|^2\times I} + f^q(w)\underbrace{\nabla^2 f(w)}_{\preceq L\times I}. \tag{4}$$

As a result, $\|\nabla^2\frac{1}{q+1}f^{q+1}(w)\|_2 \leq L_q(w) = Lf(w)^q + qf(w)^{q-1}\|\nabla f(w)\|^2$. $\qquad\square$

**A first approach: $q$-FedSGD.** Our first fair federated learning method, $q$-FedSGD, is an extension of the well-known federated mini-batch SGD (FedSGD) method (McMahan et al., 2017). $q$-FedSGD uses a dynamic step-size instead of the normal fixed step-size of FedSGD. Based on Lemma 3, for each local device $k$, the upper-bound of the local Lipschitz constant is $LF_k(w)^q + qF_k(w)^{q-1}\|\nabla F_k(w)\|^2$. In each step of $q$-FedSGD, $\nabla F_k$ and $F_k$ on each selected device $k$ are computed at the current iterate and communicated to the central node. This information is used to compute the step-sizes (weights) for combining the updates from each device. The details are summarized in Algorithm 1. Note that $q$-FedSGD is reduced to FedSGD when $q = 0$. It is also important to note that to run $q$-FedSGD with different values of $q$, we only need to estimate $L$ once by tuning the step-size on $q = 0$ and can then reuse it for all values of $q > 0$.

**Improving communication-efficiency: $q$-FedAvg.** In federated settings, communication-efficient schemes using *local* stochastic solvers (such as FedAvg) have been shown to significantly improve convergence speed (McMahan et al., 2017). However, when $q > 0$, the $F_k^{q+1}$ term is not an empirical average of the loss over all local samples due to the $q + 1$ exponent, preventing the use of local SGD as in FedAvg. To address this, we propose to generalize FedAvg for $q > 0$ using a more sophisticated dynamic weighted averaging scheme. The weights (step-sizes) are inferred from the upper bound of the local Lipschitz constants of the gradients of $F_k^{q+1}$, similar to $q$-FedSGD. To extend the local updating technique of FedAvg to the $q$-FFL objective (2), we propose a heuristic where we replace the gradient $\nabla F_k$ in the $q$-FedSGD steps with the local updates that are obtained by running SGD locally on device $k$. Similarly, $q$-FedAvg is reduced to FedAvg when $q = 0$. We

---

**Algorithm 1** $q$-`FedSGD`

---

1: **Input:** $K, T, q, 1/L, w^0, p_k, k = 1, \cdots, m$
2: **for** $t = 0, \cdots, T - 1$ **do**
3:     Server selects a subset $S_t$ of $K$ devices at random (each device $k$ is chosen with prob. $p_k$)
4:     Server sends $w^t$ to all selected devices
5:     Each selected device $k$ computes:
$$\Delta_k^t = F_k^q(w^t)\nabla F_k(w^t)$$
$$h_k^t = qF_k^{q-1}(w^t)\|\nabla F_k(w^t)\|^2 + LF_k^q(w^t)$$
6:     Each selected device $k$ sends $\Delta_k^t$ and $h_k^t$ back to the server
7:     Server updates $w^{t+1}$ as:
$$w^{t+1} = w^t - \frac{\sum_{k \in S_t} \Delta_k^t}{\sum_{k \in S_t} h_k^t}$$
8: **end for**

---

---

**Algorithm 2** $q$-`FedAvg`

---

1: **Input:** $K, E, T, q, 1/L, \eta, w^0, p_k, k = 1, \cdots, m$
2: **for** $t = 0, \cdots, T - 1$ **do**
3:     Server selects a subset $S_t$ of $K$ devices at random (each device $k$ is chosen with prob. $p_k$)
4:     Server sends $w^t$ to all selected devices
5:     Each selected device $k$ updates $w^t$ for $E$ epochs of SGD on $F_k$ with step-size $\eta$ to obtain $\bar{w}_k^{t+1}$
6:     Each selected device $k$ computes:
$$\Delta w_k^t = L(w^t - \bar{w}_k^{t+1})$$
$$\Delta_k^t = F_k^q(w^t)\Delta w_k^t$$
$$h_k^t = qF_k^{q-1}(w^t)\|\Delta w_k^t\|^2 + LF_k^q(w^t)$$
7:     Each selected device $k$ sends $\Delta_k^t$ and $h_k^t$ back to the server
8:     Server updates $w^{t+1}$ as:
$$w^{t+1} = w^t - \frac{\sum_{k \in S_t} \Delta_k^t}{\sum_{k \in S_t} h_k^t}$$
9: **end for**

---

provide additional details on $q$-`FedAvg` in Algorithm 2. As we will see empirically, $q$-`FedAvg` can solve $q$-`FFL` objective much more efficiently than $q$-`FedSGD` due to the local updating heuristic. Finally, recall that as $q \to \infty$ the $q$-`FFL` objective recovers that of the `AFL`. However, we empirically notice that $q$-`FedAvg` has a more favorable convergence speed compared to `AFL` while resulting in similar performance across devices (see Figure 9 in the appendix).

## 4 EVALUATION

We now present empirical results of the proposed objective, $q$-`FFL`, and proposed methods, $q$-`FedAvg` and $q$-`FedSGD`. We describe our experimental setup in Section 4.1. We then demonstrate the improved fairness of $q$-`FFL` in Section 4.2, and compare $q$-`FFL` with several baseline fairness objectives in Section 4.3. Finally, we show the efficiency of $q$-`FedAvg` compared with $q$-`FedSGD` in Section 4.4. All code, data, and experiments are publicly available at `github.com/litian96/fair_flearn`.

### 4.1 EXPERIMENTAL SETUP

**Federated datasets.** We explore a suite of federated datasets using both convex and non-convex models in our experiments. The datasets are curated from prior work in federated learning (McMahan et al., 2017; Smith et al., 2017; Li et al., 2020; Mohri et al., 2019) as well as recent federated learning benchmarks (Caldas et al., 2018). In particular, we study: (1) a synthetic dataset using a linear regression classifier, (2) a Vehicle dataset collected from a distributed sensor network (Duarte & Hu, 2004) with a linear SVM for binary classification, (3) tweet data curated from Sentiment140 (Go et al., 2009) (Sent140) with an LSTM classifier for text sentiment analysis, and (4) text data built

from *The Complete Works of William Shakespeare* (McMahan et al., 2017) and an RNN to predict the next character. When comparing with AFL, we use the two small benchmark datasets (Fashion MNIST (Xiao et al., 2017) and Adult (Blake, 1998)) studied in Mohri et al. (2019). When applying $q$-FFL to meta-learning, we use the common meta-learning benchmark dataset Omniglot (Lake et al., 2015). Full dataset details are given in Appendix D.1.

**Implementation.** We implement all code in Tensorflow (Abadi et al., 2016), simulating a federated network with one server and $m$ devices, where $m$ is the total number of devices in the dataset (Appendix D.1). We provide full details (including all hyperparameter values) in Appendix D.2.

## 4.2 FAIRNESS OF $q$-FFL

In our first experiments, we verify that the proposed objective $q$-FFL leads to more fair solutions (Definition 1) for federated data. In Figure 2, we compare the final testing accuracy distributions of two objectives ($q = 0$ and a tuned value of $q > 0$) averaged across 5 random shuffles of each dataset. We observe that while the average testing accuracy remains fairly consistent, the objectives with $q > 0$ result in more centered (i.e., fair) testing accuracy distributions with lower variance. In particular, *while maintaining roughly the same average accuracy, $q$-FFL reduces the variance of accuracies across all devices by 45% on average.* We further report the worst and best 10% testing accuracies and the variance of the final accuracy distributions in Table 1. Comparing $q = 0$ and $q > 0$, we see that the average testing accuracy remains almost unchanged with the proposed objective despite significant reductions in variance. We report full results on all uniformity measurements (including variance) in Table 5 in the appendix, and show that $q$-FFL encourages more uniform accuracies under other metrics as well. We observe similar results on training accuracy distributions in Figure 6 and Table 6, Appendix E. In Table 1, the average accuracy is with respect to all data points, not all devices; however, we observe similar results with respect to devices, as shown in Table 7, Appendix E.

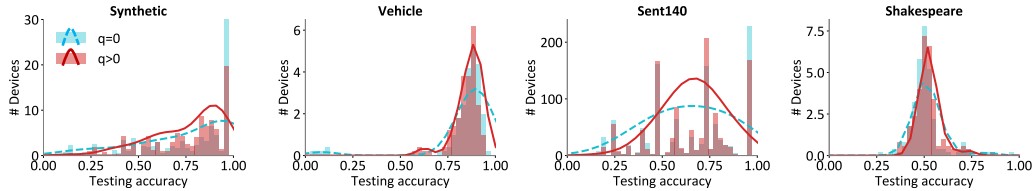

Figure 2: $q$-FFL leads to fairer test accuracy distributions. While the average accuracy remains almost identical (see Table 1), by setting $q > 0$, the distributions shift towards the center as low accuracies increase at the cost of potentially decreasing high accuracies on some devices. Setting $q = 0$ corresponds to the original objective (1). The selected $q$ values for $q > 0$ on the four datasets, as well as distribution statistics, are also shown in Table 1.

Table 1: Statistics of the test accuracy distribution for $q$-FFL. By setting $q > 0$, the accuracy of the worst 10% devices is increased at the cost of possibly decreasing the accuracy of the best 10% devices. While the average accuracy remains similar, the variance of the final accuracy distribution decreases significantly. We provide full results of other uniformity measurements (including variance) in Table 5, Appendix E.1, and show that $q$-FFL encourages more uniform distributions under all metrics.

| Dataset | Objective | Average (%) | Worst 10% (%) | Best 10% (%) | Variance |
|---|---|---|---|---|---|
| Synthetic | $q = 0$ | $80.8 \pm .9$ | $18.8 \pm 5.0$ | $100.0 \pm 0.0$ | $724 \pm 72$ |
| | $q = 1$ | $79.0 \pm 1.2$ | $\mathbf{31.1} \pm 1.8$ | $100.0 \pm 0.0$ | $\mathbf{472} \pm 14$ |
| Vehicle | $q = 0$ | $87.3 \pm .5$ | $43.0 \pm 1.0$ | $\mathbf{95.7} \pm 1.0$ | $291 \pm 18$ |
| | $q = 5$ | $87.7 \pm .7$ | $\mathbf{69.9} \pm .6$ | $94.0 \pm .9$ | $\mathbf{48} \pm 5$ |
| Sent140 | $q = 0$ | $65.1 \pm 4.8$ | $15.9 \pm 4.9$ | $100.0 \pm 0.0$ | $697 \pm 132$ |
| | $q = 1$ | $66.5 \pm .2$ | $\mathbf{23.0} \pm 1.4$ | $100.0 \pm 0.0$ | $\mathbf{509} \pm 30$ |
| Shakespeare | $q = 0$ | $51.1 \pm .3$ | $39.7 \pm 2.8$ | $\mathbf{72.9} \pm 6.7$ | $82 \pm 41$ |
| | $q = .001$ | $52.1 \pm .3$ | $\mathbf{42.1} \pm 2.1$ | $69.0 \pm 4.4$ | $\mathbf{54} \pm 27$ |

**Choosing $q$.** As discussed in Section 3.3, a natural question is to determine how $q$ should be tuned in the $q$-FFL objective. Our framework is flexible in that it allows one to choose $q$ to tradeoff between fairness/uniformity and average accuracy. We empirically show that there are a family of $q$'s that can result in variable levels of fairness (and accuracy) on synthetic data in Table 11, Appendix E. In general, this value can be tuned based on the data/application at hand and the desired amount of fairness. Another reasonable approach in practice would be to run Algorithm 2 with multiple $q$'s in parallel to obtain multiple final global models, and then select amongst these based on performance (e.g., accuracy) on the validation data. Rather than using just one optimal $q$ for all devices, for example, each device could pick a device-specific model based on their validation data. We show additional performance improvements with this device-specific strategy in Table 12 in Appendix E. Finally, we note that one potential issue is that increasing the value of $q$ may slow the speed of convergence. However, for values of $q$ that result in more fair results on our datasets, we do not observe significant decrease in the convergence speed, as shown in Figure 8, Appendix E.

### 4.3 COMPARISON WITH OTHER OBJECTIVES

Next, we compare $q$-FFL with other objectives that are likely to impose fairness in federated networks. One heuristic is to weight each data point equally, which reduces to the original objective in (1) (i.e., $q$-FFL with $q = 0$) and has been investigated in Section 4.2. We additionally compare with two alternatives: weighting *devices equally* when sampling devices, and weighting *devices adversarially*, namely, optimizing for the worst-performing device, as proposed in Mohri et al. (2019).

**Weighting devices equally.** We compare $q$-FFL with uniform sampling schemes and report testing accuracy in Figure 3. A table with the final accuracies and three fairness metrics is given in the appendix in Table 9. While the 'weighting each device equally' heuristic tends to outperform our method in *training* accuracy distributions (Figure 7 and Table 8 in Appendix E), we see that our method produces more fair solutions in terms of *testing* accuracies. One explanation for this is that uniform sampling is a static method and can easily overfit to devices with very few data points, whereas $q$-FFL will put less weight on a device once its loss becomes small, potentially providing better generalization performance due to its dynamic nature.

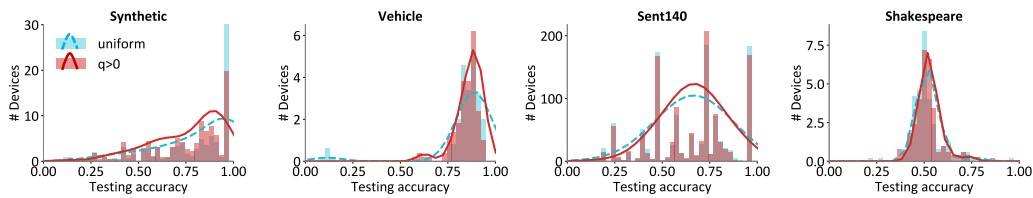

Figure 3: $q$-FFL ($q > 0$) compared with uniform sampling. In terms of testing accuracy, our objective produces more fair solutions than uniform sampling. Distribution statistics are provided in Table 9 in the appendix. $q$-FFL achieves similar average accuracies and more fair solutions.

**Weighting devices adversarially.** We further compare with AFL (Mohri et al., 2019), which is the only work we are aware of that aims to address fairness issues in federated learning. We implement a non-stochastic version of AFL where all devices are selected and updated each round, and perform grid search on the AFL hyperparameters, $\gamma_w$ and $\gamma_\lambda$. In order to devise a setup that is as favorable to AFL as possible, we modify Algorithm 2 by sampling all devices and letting each of them run gradient descent at each round. We use the same small datasets (Adult (Blake, 1998) and subsampled Fashion MNIST (Xiao et al., 2017)) and the same logistic regression model as in Mohri et al. (2019). Full details of the implementation and hyperparameters (e.g., values of $q_1$ and $q_2$) are provided in Appendix D.2.3. We note that, as opposed to AFL, $q$-FFL is flexible depending on the amount of fairness desired, with larger $q$ leading to more accuracy uniformity. As discussed, $q$-FFL generalizes AFL in this regard, as AFL is equivalent to $q$-FFL with a large enough $q$. In Table 2, we observe that $q$-FFL can in fact achieve higher testing accuracy than AFL on the device with the worst performance (i.e., the problem that the AFL was designed to solve) with appropriate $q$. This also indicates that $q$-FFL obtains the most fair solutions in certain cases. We also observe that $q$-FFL converges faster

in terms of communication rounds compared with AFL to obtain similar performance (Appendix E), which we speculate is due to the non-smoothness of the AFL objective.

Table 2: Our objective compared with weighing devices adversarially (AFL (Mohri et al., 2019)). In order to be favorable to AFL, we use the two small, well-behaved datasets studied in (Mohri et al., 2019). $q$-FFL ($q > 0$) outperforms AFL on the worst testing accuracy of both datasets. The tunable parameter $q$ controls how much fairness we would like to achieve—larger $q$ induces less variance. Each accuracy is averaged across 5 runs with different random initializations.

| | Adult | | | Fashion MNIST | | | |
|---|---|---|---|---|---|---|---|
| Objectives | average (%) | PhD (%) | non-PhD (%) | average (%) | shirt (%) | pullover (%) | T-shirt (%) |
| $q$-FFL, $q$=0 | 83.2 ±.1 | 69.9 ±.4 | **83.3** ±.1 | 78.8 ±.2 | 66.0 ±.7 | 84.5 ±.8 | **85.9** ±.7 |
| AFL | 82.5 ±.5 | 73.0 ±2.2 | 82.6 ±.5 | 77.8 ±1.2 | 71.4 ±4.2 | 81.0 ±3.6 | 82.1 ±3.9 |
| $q$-FFL, $q_1$>0 | 82.6 ±.1 | **74.1** ±.6 | 82.7 ±.1 | 77.8 ±.2 | **74.2** ±.3 | 78.9 ±.4 | 80.4 ±.6 |
| $q$-FFL, $q_2$>$q_1$ | 82.3 ±.1 | **74.4** ±.9 | 82.4 ±.1 | 77.1 ±.4 | **74.7** ±.9 | 77.9 ±.4 | 78.7 ±.6 |

## 4.4 EFFICIENCY OF THE METHOD $q$-FEDAVG

In this section, we show the efficiency of our proposed distributed solver, $q$-FedAvg, by comparing Algorithm 2 with its non-local-updating baseline $q$-FedSGD (Algorithm 1) to solve the same objective (same $q > 0$ as in Table 1). At each communication round, we have each method perform the same amount of computation, with $q$-FedAvg running one epoch of local updates on each selected device while $q$-FedSGD runs gradient descent with the local training data. In Figure 4, $q$-FedAvg converges faster than $q$-FedSGD in terms of communication rounds in most cases due to its local updating scheme. The slower convergence of $q$-FedAvg compared with $q$-FedSGD on the synthetic dataset may be due to the fact that when local data distributions are highly heterogeneous, local updating schemes may allow local models to move too far away from the initial global model, potentially hurting convergence; see Figure 10 in Appendix E for more details.

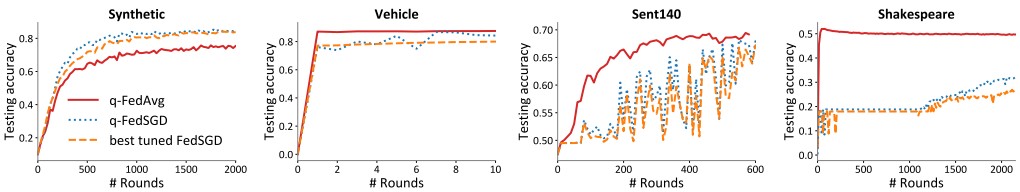

Figure 4: For a fixed objective (i.e., $q$-FFL with the same $q$), the convergence of $q$-FedAvg (Alg.2), $q$-FedSGD (Alg.1), and FedSGD. For $q$-FedAvg and $q$-FedSGD, we tune a best step-size on $q = 0$ and apply that step-size to solve $q$-FFL with $q > 0$. For $q$-FedSGD, we tune the step-size directly. We observe that (1) $q$-FedAvg converges faster in terms of communication rounds; (2) our proposed $q$-FedSGD solver with a dynamic step-size achieves similar convergence behavior compared with a best-tuned FedSGD.

To demonstrate the optimality of our dynamic step-size strategy in terms of solving $q$-FFL, we also compare our solver $q$-FedSGD with FedSGD with a best-tuned step-size. For $q$-FedSGD, we tune a step-size on $q = 0$ and apply that step-size to solve $q$-FFL with $q > 0$. $q$-FedSGD has similar performance with FedSGD, which indicates that (the inverse of) our estimated Lipschitz constant on $q > 0$ is as good as a best tuned fixed step-size. We can reuse this estimation for different $q$'s instead of manually re-tuning it when $q$ changes. We show the full results on other datasets in Appendix E. We note that both proposed methods $q$-FedAvg and $q$-FedSGD can be easily integrated into existing implementations of federated learning algorithms such as TensorFlow Federated (TFF).

### 4.5 Beyond Federated Learning: Applying $q$-FFL to Meta-learning

Finally, we generalize the proposed $q$-FFL objective to other learning tasks beyond federated learning. One natural extension is to apply $q$-FFL to meta-learning, where each task can be viewed as a device in federated networks. The goal of meta-learning is to learn a model initialization such that it can be quickly adapted to new tasks using limited training samples. However, as the new tasks can be heterogeneous, the performance distribution of the final personalized models may also be non-uniform. Therefore, we aim to learn a better initialization such that it can quickly solve unseen tasks in a fair manner, i.e., reduce the variance of the accuracy distribution of the personalized models.

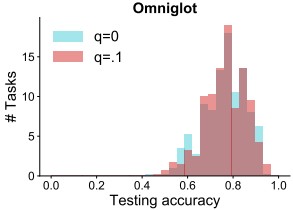

Figure 5: $q$-FFL results in fairer (more centered) initializations for meta-learning tasks.

Table 3: Statistics of the accuracy distribution of personalized models using $q$-MAML. The method with $q = 0$ corresponds to MAML. Similarly, we see that the variance is reduced while the accuracy of the worst 10% devices is increased.

| Dataset | Objective | Average (%) | Worst 10% (%) | Best 10% (%) | Variance |
|---------|-----------|-------------|---------------|--------------|----------|
| Omniglot | $q = 0$ | $79.1 \pm 9.8$ | $61.2 \pm 3.2$ | $94.0 \pm .5$ | $93 \pm 23$ |
|  | $q = .1$ | $79.3 \pm 9.6$ | $\mathbf{62.5} \pm 5.3$ | $93.8 \pm .9$ | $\mathbf{86} \pm 28$ |

To achieve this goal, we propose a new method, $q$-MAML, by combining $q$-FFL with the popular meta-learning method MAML (Finn et al., 2017). In particular, instead of updating the global model in the way described in MAML, we update the global parameters using the gradients of the $q$-FFL objective $\frac{1}{q+1} F_k^{q+1}(w)$, with weights inferred from Lemma 3. Similarly, $q$-MAML with $q = 0$ reduces to MAML, and $q$-MAML with $q \to \infty$ corresponds to MAML with a most 'fair' initialization and a potentially lower average accuracy. The detailed algorithm is summarized in Algorithm 4 in Appendix C.2. We sample 10 tasks at each round during meta-training, and train for 5 iterations of (mini-batch) SGD for personalization on meta-testing tasks. We report test accuracy of personalized models on the meta-testing tasks. From Figure 5 and Table 3 above, we observe that $q$-MAML is able to learn initializations which result in fairer personalized models with lower variance.

## 5 Conclusion

In this work, we propose $q$-FFL, a novel optimization objective inspired by fair resource allocation in wireless networks that encourages fairer (more uniform) accuracy distributions across devices in federated learning. We devise a scalable method, $q$-FedAvg, to solve this objective in massive networks. Our empirical evaluation on a suite of federated datasets demonstrates the resulting fairness and flexibility of $q$-FFL, as well as the efficiency of $q$-FedAvg compared with existing baselines. We show that our framework is useful not only for federated learning tasks, but also for other learning paradigms such as meta-learning.

## Acknowledgments

We thank Sebastian Caldas, Chen Dan, Neel Guha, Anit Kumar Sahu, Eric Tan, and Samuel Yeom for their helpful discussions and comments. The work of TL and VS was supported in part by the National Science Foundation grant IIS1838017, a Google Faculty Award, a Carnegie Bosch Institute Research Award, and the CONIX Research Center. Any opinions, findings, and conclusions or recommendations expressed in this material are those of the author(s) and do not necessarily reflect the National Science Foundation or any other funding agency.

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

## A    THEORETICAL ANALYSIS OF THE PROPOSED OBJECTIVE $q$-FFL

### A.1    UNIFORMITY INDUCED BY $q$-FFL

In this section, we theoretically justify that the $q$-FFL objective can impose more uniformity of the performance/accuracy distribution. As discussed in Section 3.2, $q$-FFL can encourage more fair solutions in terms of several metrics, including (1) the variance of accuracy distribution (smaller variance), (2) the cosine similarity between the accuracy distribution and the all-ones vector $\mathbf{1}$ (larger similarity), and (3) the entropy of the accuracy distribution (larger entropy). We begin by formally defining these fairness notions.

**Definition 4** (*Uniformity 1: Variance of the performance distribution*)**.** We say that the performance distribution of $m$ devices $\{F_1(w), \ldots, F_m(w)\}$ is more uniform under solution $w$ than $w'$ if

$$\mathbf{Var}(F_1(w), \ldots, F_m(w)) < \mathbf{Var}(F_1(w'), \ldots, F_m(w')). \tag{5}$$

**Definition 5** (*Uniformity 2: Cosine similarity between the performance distribution and* $\mathbf{1}$)**.** We say that the performance distribution of $m$ devices $\{F_1(w), \ldots, F_m(w)\}$ is more uniform under solution $w$ than $w'$ if the cosine similarity between $\{F_1(w), \ldots, F_m(w)\}$ and $\mathbf{1}$ is larger than that between $\{F_1(w'), \ldots, F_m(w')\}$ and $\mathbf{1}$, i.e.,

$$\frac{\frac{1}{m} \sum_{k=1}^{m} F_k(w)}{\sqrt{\frac{1}{m} \sum_{k=1}^{m} F_k^2(w)}} \geq \frac{\frac{1}{m} \sum_{k=1}^{m} F_k(w')}{\sqrt{\frac{1}{m} \sum_{k=1}^{m} F_k^2(w')}}. \tag{6}$$

**Definition 6** (*Uniformity 3: Entropy of performance distribution*)**.** We say that the performance distribution of $m$ devices $\{F_1(w), \ldots, F_m(w)\}$ is more uniform under solution $w$ than $w'$ if

$$\widetilde{H}(F(w)) \geq \widetilde{H}(F(w')), \tag{7}$$

where $\widetilde{H}(F(w))$ is the entropy of the stochastic vector obtained by normalizing $\{F_1(w), \ldots, F_m(w)\}$, defined as

$$\widetilde{H}(F(w)) := -\sum_{k=1}^{m} \frac{F_k(w)}{\sum_{k=1}^{m} F_k(w)} \log\left(\frac{F_k(w)}{\sum_{k=1}^{m} F_k(w)}\right). \tag{8}$$

To enforce uniformity/fairness (defined in Definition 4, 5, and 6), we propose the $q$-FFL objective to impose more weights on the devices with worse performance. Throughout the proof, for the ease of mathematical exposition, we consider a similar unweighted objective:

$$\min_{w} \left\{ f_q(w) = \left(\frac{1}{m} \sum_{k=1}^{m} F_k^{q+1}(w)\right)^{\frac{1}{q+1}} \right\},$$

and we denote $w_q^*$ as the global optimal solution of $\min_w \ f_q(w)$.

We first investigate the special case of $q = 1$ and show that $q = 1$ results in more fair solutions than $q = 0$ based on Definition 4 and Definition 5.

**Lemma 7.** $q = 1$ *leads to a more fair solution (smaller variance of the model performance distribution) than* $q = 0$, *i.e.,* $\mathbf{Var}(F_1(w_1^*), \ldots, F_m(w_1^*)) < \mathbf{Var}(F_1(w_0^*), \ldots, F_m(w_0^*))$.

*Proof.* Use the fact that $w_1^*$ is the optimal solution of $\min_w \ f_1(w)$, and $w_0^*$ is the optimal solution of $\min_w \ f_0(w)$, we get

$$\frac{\sum_{k=1}^{m} F_k^2(w_1^*)}{m} - \left(\frac{1}{m} \sum_{i=1}^{m} F_k(w_1^*)\right)^2 \leq \frac{\sum_{k=1}^{m} F_k^2(w_0^*)}{m} - \left(\frac{1}{m} \sum_{i=1}^{m} F_k(w_1^*)\right)^2$$

$$\leq \frac{\sum_{k=1}^{m} F_k^2(w_0^*)}{m} - \left(\frac{1}{m} \sum_{i=1}^{m} F_k(w_0^*)\right)^2. \tag{9}$$

$\square$

**Lemma 8.** $q = 1$ *leads to a more fair solution (larger cosine similarity between the performance distribution and* **1***) than* $q = 0$*, i.e.,*

$$\frac{\frac{1}{m}\sum_{k=1}^{m} F_k(w_1^*)}{\sqrt{\frac{1}{m} F_k^2(w_1^*)}} \geq \frac{\frac{1}{m}\sum_{k=1}^{m} F_k(w_0^*)}{\sqrt{\frac{1}{m} F_k^2(w_0^*)}}.$$

*Proof.* As $\frac{1}{m}\sum_{k=1}^{m} F_k(w_1^*) \geq \frac{1}{m}\sum_{k=1}^{m} F_k(w_0^*)$ and $\frac{1}{m}\sum_{k=1}^{m} F_k^2(w_1^*) \geq \frac{1}{m}\sum_{k=1}^{m} F_k^2(w_0^*)$, it directly follows that

$$\frac{\frac{1}{m}\sum_{k=1}^{m} F_k(w_1^*)}{\sqrt{\frac{1}{m} F_k^2(w_1^*)}} \geq \frac{\frac{1}{m}\sum_{k=1}^{m} F_k(w_0^*)}{\sqrt{\frac{1}{m} F_k^2(w_0^*)}}.$$

$\square$

We next provide results based on Definition 6. It states that for arbitrary $q \geq 0$, by increasing $q$ for a small amount, we can get more uniform performance distributions defined over higher-orders of the performance.

**Lemma 9.** *Let* $F(w)$ *be twice differentiable in* $w$ *with* $\nabla^2 F(w) \succ 0$ *(positive definite). The derivative of* $\widetilde{H}(F^q(w_p^*))$ *with respect to the variable* $p$ *evaluated at the point* $p = q$ *is non-negative, i.e.,*

$$\left.\frac{\partial}{\partial p}\widetilde{H}(F^q(w_p^*))\right|_{p=q} \geq 0,$$

*where* $\widetilde{H}(F^q(w_p^*))$ *is defined in equation 8.*

*Proof.*

$$\left.\frac{\partial}{\partial p}\widetilde{H}(F^q(w_p^*))\right|_{p=q} = -\left.\frac{\partial}{\partial p}\sum_k \frac{F_k^q(w_p^*)}{\sum_\kappa F_\kappa^q(w_p^*)} \ln\left(\frac{F_k^q(w_p^*)}{\sum_\kappa F_\kappa^q(w_p^*)}\right)\right|_{p=q} \tag{10}$$

$$= -\left.\frac{\partial}{\partial p}\sum_k \frac{F_k^q(w_p^*)}{\sum_\kappa F_\kappa^q(w_p^*)} \ln\left(F_k^q(w_p^*)\right)\right|_{p=q}$$

$$+ \left.\frac{\partial}{\partial p}\ln\sum_\kappa F_\kappa^q(w_p^*)\right|_{p=q} \tag{11}$$

$$= -\sum_k \frac{\left(\left.\frac{\partial}{\partial p}w_p^*\right|_{p=q}\right)^\top \nabla_w F_k^q(w_q^*)}{\sum_\kappa F_\kappa^q(w_q^*)} \ln\left(F_k^q(w_q^*)\right)$$

$$- \sum_k \frac{F_k^q(w_q^*)}{\sum_\kappa F_\kappa^q(w_q^*)} \frac{\left(\left.\frac{\partial}{\partial p}w_p^*\right|_{p=q}\right)^\top \nabla_w F_k^q(w_q^*)}{F_k^q(w_q^*)} \tag{12}$$

$$= -\sum_k \frac{\left(\left.\frac{\partial}{\partial p}w_p^*\right|_{p=q}\right)^\top \nabla_w F_k^q(w_q^*)}{\sum_\kappa F_\kappa^q(w_q^*)} \left(\ln\left(F_k^q(w_q^*)\right) + 1\right). \tag{13}$$

Now, let us examine $\left.\frac{\partial}{\partial p}w_p^*\right|_{p=q}$. We know that $\sum_k \nabla_w F_k^p(w_p^*) = 0$ by definition. Taking the derivative with respect to $p$, we have

$$\sum_k \nabla_w^2 F_k^p(w_p^*)\frac{\partial}{\partial p}w_p^* + \sum_k \left(\ln F_k^p(w_p^*) + 1\right)\nabla_w F_k^p(w_p^*) = 0. \tag{14}$$

Invoking implicit function theorem,

$$\frac{\partial}{\partial p}w_p^* = -\left(\sum_k \nabla_w^2 F_k^p(w_p^*)\right)^{-1}\sum_k \left(\ln F_k^p(w_p^*) + 1\right)\nabla_w F_k^p(w_p^*). \tag{15}$$

Plugging $\frac{\partial}{\partial p} w_p^* \big|_{p=q}$ into (13), we get that $\frac{\partial}{\partial p} \widetilde{H}(F^q(w_p^*)) \big|_{p=q} \geq 0$ completing the proof. □

Lemma 9 states that for any $p$, the performance distribution of $\{F_1^p(w_{p+\epsilon}^*), \ldots, F_m^p(w_{p+\epsilon}^*)\}$ is guaranteed to be more uniform based on Definition 6 than that of $\{F_1^p(w_p^*), \ldots, F_m^p(w_p^*)\}$ for a small enough $\epsilon$. Note that Lemma 9 is different from the existing results on the monotonicity of entropy under the tilt operation, which would imply that $\frac{\partial}{\partial q} \widetilde{H}(F^q(w_p^*)) \leq 0$ for all $q \geq 0$ (see Beirami et al. (2019, Lemma 11)).

Ideally, we would like to prove a result more general than Lemma 9, implying that the distribution $\{F_1^q(w_{p+\epsilon}^*), \ldots, F_m^q(w_{p+\epsilon}^*)\}$ is more uniform than $\{F_1^q(w_p^*), \ldots, F_m^q(w_p^*)\}$ for any $p, q$ and small enough $\epsilon$. We prove this result for the special case of $m = 2$ in the following.

**Lemma 10.** *Let $F(w)$ be twice differentiable in $w$ with $\nabla^2 F(w) \succ 0$ (positive definite). If $m = 2$, for any $q \in \mathbb{R}^+$, the derivative of $\widetilde{H}(F^q(w_p^*))$ with respect to the variable $p$ is non-negative, i.e.,*

$$\frac{\partial}{\partial p} \widetilde{H}(F^q(w_p^*)) \geq 0,$$

*where $\widetilde{H}(F^q(w_p^*))$ is defined in equation 8.*

*Proof.* First, we invoke Lemma 9 to obtain that

$$\frac{\partial}{\partial p} \widetilde{H}(F^q(w_p^*)) \bigg|_{p=q} \geq 0. \tag{16}$$

Let

$$\theta_q(w) := \frac{F_1^q(w)}{F_1^q(w) + F_2^q(w)}. \tag{17}$$

Without loss of generality assume that $\theta_q(w_p^*) \in (0, \frac{1}{2}]$, as we can relabel $F_1$ and $F_2$ otherwise. Then, given that $m = 2$, we conclude from equation 16 along with the monotonicity of the binary entropy function in $(0, \frac{1}{2}]$ that

$$\frac{\partial}{\partial p} \theta_q(w_p^*) \bigg|_{p=q} \geq 0, \tag{18}$$

which in conjunction with equation 17 implies that

$$\frac{\partial}{\partial p} \left( \frac{F_1(w_p^*)}{F_2(w_p^*)} \right)^q \bigg|_{p=q} \geq 0. \tag{19}$$

Given the monotonicity of $x^q$ with respect to $x$ for all $q > 0$, it can be observed that the above is sufficient to imply that for any $q > 0$,

$$\frac{\partial}{\partial p} \left( \frac{F_1(w_p^*)}{F_2(w_p^*)} \right)^q \geq 0. \tag{20}$$

Going all of the steps back we would obtain that for all $p > 0$

$$\frac{\partial}{\partial p} \widetilde{H}(F^q(w_p^*)) \geq 0. \tag{21}$$

This completes the proof of the lemma. □

We conjecture that the statement of Lemma 9 is true for all $q \in \mathbb{R}^+$, which would be equivalent to the statement of Lemma 10 holding true for all $m \in \mathbb{N}$.

Thus far, we provided results that showed that $q$-FFL promotes fairness in three different senses. Next, we further provide a result on equivalence between the geometric and information-theoretic notions of fairness.

**Lemma 11** (*Equivalence between uniformity in entropy and cosine distance*). *$q$-FFL encourages more uniform performance distributions in the cosine distance sense (Definition 5) if any only if it encourages more uniform performance distributions in the entropy sense (Definition 6), i.e., (a) holds if and only if (b) holds where*
*(a) for any $p, q \in \mathbb{R}$, the derivative of $H(F^q(w_p^*))$ with respect to $p$ is non-negative,*
*(b) for any $0 \le t \le r, 0 \le v \le u$, $\frac{f_t(w_u^*)}{f_r(w_u^*)} \ge \frac{f_t(w_v^*)}{f_r(w_v^*)}$.*

*Proof.* Definition 6 is a special case of $H(F^q(w_p^*))$ with $q = 1$. If $\widetilde{H}(F^q(w_p^*))$ increases with $p$ for any $p, q$, then we are guaranteed to get more fair solutions based on Definition 6. Similarly, Definition 5 is a special case of $\frac{f_t(w_u^*)}{f_r(w_u^*)}$ with $t = 0, r = 1$. If $\frac{f_t(w_u^*)}{f_r(w_u^*)}$ increases with $u$ for any $t \le r$, $q$-FFL can also obtain more fair solutions under Definition 5.

Next, we show that (a) and (b) are equivalent measures of fairness.

For any $r \ge t \ge 0$, and any $u \ge v \ge 0$,

$$\frac{f_t(w_u^*)}{f_r(w_u^*)} \ge \frac{f_t(w_v^*)}{f_r(w_v^*)} \iff \ln \frac{f_t(w_u^*)}{f_r(w_u^*)} - \ln \frac{f_t(w_v^*)}{f_r(w_v^*)} \ge 0 \tag{22}$$

$$\iff \int_v^u \frac{\partial}{\partial \tau} \ln \frac{f_t(w_\tau^*)}{f_r(w_\tau^*)} d\tau \ge 0 \tag{23}$$

$$\iff \frac{\partial}{\partial p} \ln \frac{f_t(w_p^*)}{f_r(w_p^*)} \ge 0, \text{ for any } p \ge 0 \tag{24}$$

$$\iff \frac{\partial}{\partial p} \ln f_r(w_p^*) - \frac{\partial}{\partial p} \ln f_t(w_p^*) \le 0, \text{ for any } p \ge 0 \tag{25}$$

$$\iff \int_t^r \frac{\partial^2}{\partial p \partial q} \ln f_q(w_p^*) dq \le 0 \text{ for any } p, q \ge 0 \tag{26}$$

$$\iff \frac{\partial^2}{\partial p \partial q} \ln f_q(w_p^*) \le 0, \text{ for any } p, q \ge 0 \tag{27}$$

$$\iff \frac{\partial}{\partial p} \widetilde{H}(F^q(w_p^*)) \ge 0, \text{ for any } p, q \ge 0. \tag{28}$$

The last inequality is obtained using the fact that by taking the derivative of $\ln f_q(w_p^*)$ with respect to $q$, we get $-\widetilde{H}(F^q(w_p^*))$. $\qquad \square$

**Discussions.** We give geometric (Definition 5) and information-theoretic (Definition 6) interpretations of our uniformity/fairness notion and provide uniformity guarantees under the $q$-FFL objective in some cases (Lemma 7, Lemma 8, and Lemma 9). We reveal interesting relations between the geometric and information-theoretic interpretations in Lemma 11. Future work would be to gain further understandings for more general cases indicated in Lemma 11.

### A.2 GENERALIZATION BOUNDS

In this section, we first describe the setup we consider in more detail, and then provide generalization bounds of $q$-FFL. One benefit of $q$-FFL is that it allows for a flexible trade-off between fairness and accuracy, which generalizes AFL (a special case of $q$-FFL with $q \to \infty$). We also provide learning bounds that generalize the bounds of the AFL objective, as described below.

Suppose the service provider is interested in minimizing the loss over a distributed network of devices, with possibly unknown weights on each device:

$$L_\lambda(h) = \sum_{k=1}^m \lambda_k \mathbb{E}_{(x,y) \sim D_k}[l(h(x), y)], \tag{29}$$

where $\lambda$ is in a probability simplex $\Lambda$, $m$ is the total number of devices, $D_k$ is the local data distribution for device $k$, $h$ is the hypothesis function, and $l$ is the loss. We use $\hat{L}_\lambda(h)$ to denote the empirical

loss:

$$\hat{L}_\lambda(h) = \sum_{k=1}^{m} \frac{\lambda_k}{n_k} \sum_{j=1}^{n_k} l(h(x_{k,j}), y_{k,j}), \tag{30}$$

where $n_k$ is the number of local samples on device $k$ and $(x_{k,j}, y_{k,j}) \sim D_k$.

We consider a slightly different, unweighted version of $q$-FFL:

$$\min_w \ f_q(w) = \frac{1}{m} \sum_{k=1}^{m} F_k^{q+1}(w), \tag{31}$$

which is equivalent to minimizing the empirical loss

$$\tilde{L}_q(h) = \max_{\nu, \|\nu\|_p \leq 1} \sum_{k=1}^{m} \frac{\nu_i}{n_k} \sum_{j=1}^{n_k} l(h(x_{k,j}), y_{k,j}), \tag{32}$$

where $\frac{1}{p} + \frac{1}{q+1} = 1$ ($p \geq 1, q \geq 0$).

**Lemma 12** (*Generalization bounds of $q$-FFL for a specific $\lambda$*)**.** *Assume that the loss $l$ is bounded by $M > 0$ and the numbers of local samples are $(n_1, \cdots, n_m)$. Then, for any $\delta > 0$, with probability at least $1 - \delta$, the following holds for any $\lambda \in \Lambda, h \in H$:*

$$L_\lambda(h) \leq A_q(\lambda)\tilde{L}_q(h) + \mathbb{E}\left[\max_{h \in H} L_\lambda(h) - \hat{L}_\lambda(h)\right] + M\sqrt{\sum_{k=1}^{m} \frac{\lambda_k^2}{2n_k} \log \frac{1}{\delta}}, \tag{33}$$

*where $A_q(\lambda) = \|\lambda\|_p$, and $1/p + 1/(q+1) = 1$.*

*Proof.* Similar to the proof in Mohri et al. (2019), for any $\delta > 0$, the following inequality holds with probability at least $1 - \delta$ for any $\lambda \in \Lambda, h \in H$:

$$L_\lambda(h) \leq \hat{L}_\lambda(h) + \mathbb{E}\left[\max_{h \in H} L_\lambda(h) - \hat{L}_\lambda(h)\right] + M\sqrt{\sum_{k=1}^{m} \frac{\lambda_k^2}{2n_k} \log \frac{1}{\delta}}. \tag{34}$$

Denote the empirical loss on device $k$ $\frac{1}{n_k} \sum_{j=1}^{n_k} l(h(x_{k,j}), y_{k,j})$ as $F_k$. From Hölder's inequality, we have

$$\hat{L}_\lambda(h) = \sum_{k=1}^{m} \lambda_k F_k \leq \left(\sum_{k=1}^{m} \lambda_k^p\right)^{\frac{1}{p}} \left(\sum_{k=1}^{m} F_k^{q+1}\right)^{\frac{1}{q+1}} = A_q(\lambda)\tilde{L}_q(h), \ \frac{1}{p} + \frac{1}{q+1} = 1.$$

Plugging $\hat{L}_\lambda(h) \leq A_q(\lambda)\tilde{L}_q(h)$ into (34) yields the results. $\qquad\qquad\square$

**Theorem 13** (*Generalization bounds of $q$-FFL for any $\lambda$*)**.** *Assume that the loss $l$ is bounded by $M > 0$ and the number of local samples is $(n_1, \cdots, n_m)$. Then, for any $\delta > 0$, with probability at least $1 - \delta$, the following holds for any $\lambda \in \Lambda, h \in H$:*

$$L_\lambda(h) \leq \max_{\lambda \in \Lambda} (A_q(\lambda)) \tilde{L}_q(h) + \max_{\lambda \in \Lambda} \left(\mathbb{E}\left[\max_{h \in H} L_\lambda(h) - \hat{L}_\lambda(h)\right] + M\sqrt{\sum_{k=1}^{m} \frac{\lambda_k^2}{2n_k} \log \frac{1}{\delta}}\right), \tag{35}$$

*where $A_q(\lambda) = \|\lambda\|_p$, and $1/p + 1/(q+1) = 1$.*

*Proof.* This directly follows from Lemma 12, by taking the maximum over all possible $\lambda$'s in $\Lambda$. $\quad\square$

**Discussions.** From Lemma 12, letting $\lambda = \left(\frac{1}{m}, \cdots, \frac{1}{m}\right)$ and $q \to \infty$, we recover the generalization bounds in AFL (Mohri et al., 2019). In that sense, our generalization results extend those of AFL's. In addition, it is not straightforward to derive an optimal $q$ with the tightest generalization bound from Lemma 12 and Theorem 13. In practice, our proposed method $q$-FedAvg allows us to tune a family of $q$'s by re-using the step-sizes.

## B $\alpha$-FAIRNESS AND $q$-FFL

As discussed in Section 2, while it is natural to consider the $\alpha$-fairness framework for machine learning, we are unaware of any work that uses $\alpha$-fairness to modify machine learning training objectives. We provide additional details on the framework below; for further background on $\alpha$-fairness and fairness in resource allocation more generally, we defer the reader to Shi et al. (2014); Mo & Walrand (2000).

$\alpha$-fairness (Lan et al., 2010; Mo & Walrand, 2000) is a popular fairness metric widely-used in resource allocation problems. The framework defines a family of overall utility functions that can be derived by summing up the following function of the individual utilities of the users in the network:

$$U_\alpha(x) = \begin{cases} \ln(x), & \text{if } \alpha = 1 \\ \frac{1}{1-\alpha} x^{1-\alpha}, & \text{if } \alpha \geq 0, \alpha \neq 1 \,. \end{cases}$$

Here $U_\alpha(x)$ represents the individual utility of some specific user given $x$ allocated resources (e.g., bandwidth). The goal is to find a resource allocation strategy to maximize the sum of the individual utilities. This family of functions includes a wide range of popular fair resource allocation strategies. In particular, the above function represents zero fairness with $\alpha = 0$, proportional fairness (Kelly, 1997) with $\alpha = 1$, harmonic mean fairness (Dashti et al., 2013) with $\alpha = 2$, and max-min fairness (Radunovic & Le Boudec, 2007) with $\alpha = +\infty$.

Note that in federated learning, we are dealing with costs and not utilities. Thus, max-min in resource allocation corresponds to min-max in our setting. With this analogy, it is clear that in our proposed objective $q$-FFL (2), the case where $q = +\infty$ corresponds to min-max fairness since it is optimizing for the worst-performing device, similar to what was proposed in Mohri et al. (2019). Also, $q = 0$ corresponds to zero fairness, which reduces to the original FedAvg objective (1). In resource allocation problems, $\alpha$ can be tuned for trade-offs between fairness and system efficiency. In federated settings, $q$ can be tuned based on the desired level of fairness (e.g., desired variance of accuracy distributions) and other performance metrics such as the overall accuracy. For instance, in Table 2 in Section 4.3, we demonstrate on two datasets that as $q$ increases, the overall average accuracy decreases slightly while the worst accuracies are increased significantly and the variance of the accuracy distribution decreases.

## C  PSEUDO-CODE OF ALGORITHMS

### C.1  THE FEDAVG ALGORITHM

---

**Algorithm 3** Federated Averaging McMahan et al. (2017) (FedAvg)

---

**Input:** $K, T, \eta, E, w^0, N, p_k, k = 1, \cdots, N$
**for** $t = 0, \cdots, T - 1$ **do**
    Server randomly chooses a subset $S_t$ of $K$ devices (each device $k$ is chosen with probability $p_k$)
    Server sends $w^t$ to all chosen devices
    Each device $k$ updates $w^t$ for $E$ epochs of SGD on $F_k$ with step-size $\eta$ to obtain $w_k^{t+1}$
    Each chosen device $k$ sends $w_k^{t+1}$ back to the server
    Server aggregates the $w$'s as $w^{t+1} = \frac{1}{K} \sum_{k \in S_t} w_k^{t+1}$
**end for**

---

### C.2  THE $q$-MAML ALGORITHM

---

**Algorithm 4** $q$-FFL applied to MAML: $q$-MAML

---

1: **Input:** $K, T, \eta, w^0, N, p_k, k = 1, \cdots, N$
2: **for** $t = 0, \cdots, T - 1$ **do**
3:     Sample a batch of $S_t$ ($|S_t| = K$) tasks randomly (each task $k$ is chosen with probability $p_k$)
4:     Send $w^t$ to all sampled tasks
5:     Each task $k \in S_t$ samples data $D_k$ from the training set and $D_k'$ from the testing set, and computes updated parameters on $D$: $w_k^t = w^t - \eta \nabla F_k(w^t)$
6:     Each task $k \in S_t$ computes the gradients $\nabla F_k(w_k^t)$ on $D'$
7:     Each task $k \in S_t$ computes:

$$\Delta_k^t = F_k^q(w_k^t) \nabla F_k(w_k^t)$$
$$h_k^t = q F_k^{q-1}(w_k^t) \|\nabla F_k(w_k^t)\|^2 + L F_k^q(w_k^t)$$

8:     $w^{t+1}$ is updated as:

$$w^{t+1} = w^t - \frac{\sum_{k \in S_t} \Delta_k^t}{\sum_{k \in S_t} h_k^t}$$

9: **end for**

---

# D    EXPERIMENTAL DETAILS

## D.1    DATASETS AND MODELS

We provide full details on the datasets and models used in our experiments. The statistics of four federated datasets used in federated learning (as opposed to meta-learning) experiments are summarized in Table 4. We report the total number of devices, the total number of samples, and mean and deviation in the sizes of total data points on each device. Additional details on the datasets and models are described below.

- **Synthetic:**   We follow a similar set up as that in Shamir et al. (2014) and impose additional heterogeneity. The model is $y = argmax(\text{softmax}(Wx+b))$, $x \in \mathbb{R}^{60}, W \in \mathbb{R}^{10 \times 60}, b \in \mathbb{R}^{10}$, and the goal is to learn a global $W$ and $b$. Samples $(X_k, Y_k)$ and local models on each device $k$ satisfies $W_k \sim \mathcal{N}(u_k, 1), b_k \sim \mathcal{N}(u_k, 1), u_k \sim \mathcal{N}(0, 1); x_k \sim \mathcal{N}(v_k, \Sigma)$, where the covariance matrix $\Sigma$ is diagonal with $\Sigma_{j,j} = j^{-1.2}$. Each element in $v_k$ is drawn from $\mathcal{N}(B_k, 1), B_k \sim \mathcal{N}(0, 1)$. There are 100 devices in total and the number of samples on each devices follows a power law.

- **Vehicle[2]:** We use the same Vehicle Sensor (Vehicle) dataset as Smith et al. (2017), modelling each sensor as a device. This dataset consists of acoustic, seismic, and infrared sensor data collected from a distributed network of 23 sensors Duarte & Hu (2004). Each sample has a 100-dimension feature and a binary label. We train a linear SVM to predict between AAV-type and DW-type vehicles. We tune the hyperparameters in SVM and report the best configuration.

- **Sent140:**   This dataset is a collection of tweets curated from 1,101 accounts from Senti-ment140 (Go et al., 2009) (Sent140) where each Twitter account corresponds to a device. The task is text sentiment analysis which we model as a binary classification problem. The model takes as input a 25-word sequence, embeds each word into a 300-dimensional space using pre-trained Glove (Pennington et al., 2014), and outputs a binary label after two LSTM layers and one densely-connected layer.

- **Shakespeare:**   This dataset is built from *The Complete Works of William Shakespeare* (McMahan et al., 2017). Each speaking role in the plays is associated with a device. We subsample 31 speaking roles to train a deep language model for next character prediction. The model takes as input an 80-character sequence, embeds each character into a learnt 8-dimensional space, and outputs one character after two LSTM layers and one densely-connected layer.

- **Omniglot:**   The Omniglot dataset (Lake et al., 2015) consists of 1,623 characters from 50 different alphabets. We create 300 meta-training tasks from the first 1,200 characters, and 100 meta-testing tasks from the last 423 characters. Each task is a 5-class classification problem where each character forms a class. The model is a convolutional neural network with two convolution layers and two fully-connected layers.

Table 4: Statistics of federated datasets

| Dataset | Devices | Samples | Samples/device | |
|---|---|---|---|---|
| | | | mean | stdev |
| Synthetic | 100 | 12,697 | 127 | 73 |
| Vehicle | 23 | 43,695 | 1,899 | 349 |
| Sent140 | 1,101 | 58,170 | 53 | 32 |
| Shakespeare | 31 | 116,214 | 3,749 | 6,912 |

## D.2    IMPLEMENTATION DETAILS

### D.2.1    MACHINES

We simulate the federated setting (one server and $m$ devices) on a server with 2 Intel® Xeon® E5-2650 v4 CPUs and 8 NVidia® 1080Ti GPUs.

---

[2]http://www.ecs.umass.edu/~mduarte/Software.html

### D.2.2 SOFTWARE

We implement all code in TensorFlow (Abadi et al., 2016) Version 1.10.1. Please see `github.com/litian96/fair_flearn` for full details.

### D.2.3 HYPERPARAMETERS

We randomly split data on each local device into 80% training set, 10% testing set, and 10% validation set. We tune a best $q$ from $\{0.001, 0.01, 0.1, 0.5, 1, 2, 5, 10, 15\}$ on the validation set and report accuracy distributions on the testing set. We pick up the $q$ value where the variance decreases the most, while the overall average accuracy change (compared with the $q = 0$ case) is within 1%. For each dataset, we repeat this process for five randomly selected train/test/validation splits, and report the mean and standard deviation across these five runs where applicable. For Synthetic, Vehicle, Sent140, and Shakespeare, optimal $q$ values are 1, 5, 1, and 0.001, respectively. For all datasets, we randomly sample 10 devices each round. We tune the learning rate and batch size on `FedAvg` and use the same learning rate and batch size for all $q$-`FedAvg` experiments of that dataset. The learning rates for Synthetic, Vehicle, Sent140, and Shakespeare are 0.1, 0.01, 0.03, and 0.8, respectively. The batch sizes for Synthetic, Vehicle, Sent140, and Shakespeare are 10, 64, 32, and 10. The number of local epochs $E$ is fixed to be 1 for both `FedAvg` and $q$-`FedAvg` regardless of the values of $q$.

In comparing $q$-`FedAvg`'s efficiency with $q$-`FedSGD`, we also tune a best learning rate for $q$-`FedSGD` methods on $q = 0$. For each comparison, we fix devices selected and mini-batch orders across all runs. We stop training when the training loss $F(w)$ does not decrease for 10 rounds. When running `AFL` methods, we search for a best $\gamma_w$ and $\gamma_\lambda$ such that `AFL` achieves the highest testing accuracy on the device with the highest loss within a fixed number of rounds. For Adult, we use $\gamma_w = 0.1$ and $\gamma_\lambda = 0.1$; for Fashion MNIST, we use $\gamma_w = 0.001$ and $\gamma_\lambda = 0.01$. We use the same $\gamma_w$ as step-sizes for $q$-`FedAvg` on Adult and Fashion MNIST. In Table 2, $q_1 = 0.01, q_2 = 2$ for $q$-`FFL` on Adult and $q_1 = 5, q_2 = 15$ for $q$-`FFL` on Fashion MNIST. Similarly, the number of local epochs is fixed to 1 whenever we perform local updates.

# E    FULL EXPERIMENTS

## E.1    FULL RESULTS OF PREVIOUS EXPERIMENTS

**Fairness of $q$-FFL under all uniformity metrics.**    We demonstrate the fairness of $q$-FFL in Table 1 in terms of variance. Here, we report similar results in terms of other uniformity measures (the last two columns).

Table 5: Full statistics of the test accuracy distribution for $q$-FFL. $q$-FFL increases the accuracy of the worst 10% devices without decreasing the average accuracies. We see that $q$-FFL encourages more uniform distributions under all uniformity metrics defined in Appendix A.2: (1) the variance of the accuracy distribution (Definition 4), (2) the cosine similarity/geometric angle between the accuracy distribution and the all-ones vector $\mathbf{1}$ (Definition 5), and (3) the KL-divergence between the normalized accuracy vector $a$ and the uniform distribution $u$, which can be directly translated to the entropy of $a$ (Definition 6) .

| Dataset | Objective | Average (%) | Worst 10% (%) | Best 10% (%) | Variance | Angle (°) | KL($a\|\|u$) |
|---|---|---|---|---|---|---|---|
| Synthetic | $q = 0$ | $80.8 \pm .9$ | $18.8 \pm 5.0$ | $100.0 \pm 0.0$ | $724 \pm 72$ | $19.5 \pm 1.1$ | $.083 \pm .013$ |
| | $q = 1$ | $79.0 \pm 1.2$ | $\mathbf{31.1} \pm 1.8$ | $100.0 \pm 0.0$ | $\mathbf{472} \pm 14$ | $\mathbf{16.0} \pm .5$ | $\mathbf{.049} \pm .003$ |
| Vehicle | $q = 0$ | $87.3 \pm .5$ | $43.0 \pm 1.0$ | $\mathbf{95.7} \pm 1.0$ | $291 \pm 18$ | $11.3 \pm .3$ | $.031 \pm .003$ |
| | $q = 5$ | $87.7 \pm .7$ | $\mathbf{69.9} \pm .6$ | $94.0 \pm .9$ | $\mathbf{48} \pm 5$ | $\mathbf{4.6} \pm .2$ | $\mathbf{.003} \pm .000$ |
| Sent140 | $q = 0$ | $65.1 \pm 4.8$ | $15.9 \pm 4.9$ | $100.0 \pm 0.0$ | $697 \pm 132$ | $22.4 \pm 3.3$ | $.104 \pm .034$ |
| | $q = 1$ | $66.5 \pm .2$ | $\mathbf{23.0} \pm 1.4$ | $100.0 \pm 0.0$ | $\mathbf{509} \pm 30$ | $\mathbf{18.8} \pm .5$ | $\mathbf{.067} \pm .006$ |
| Shakespeare | $q = 0$ | $51.1 \pm .3$ | $39.7 \pm 2.8$ | $72.9 \pm 6.7$ | $82 \pm 41$ | $9.8 \pm 2.7$ | $.014 \pm .006$ |
| | $q = .001$ | $52.1 \pm .3$ | $\mathbf{42.1} \pm 2.1$ | $69.0 \pm 4.4$ | $\mathbf{54} \pm 27$ | $\mathbf{7.9} \pm 2.3$ | $\mathbf{.009} \pm .05$ |

**Fairness of $q$-FFL with respect to training accuracy.**    The empirical results in Section 4 are with respect to testing accuracy. As a sanity check, we show that $q$-FFL also results in more fair training accuracy distributions in Figure 6 and Table 6.

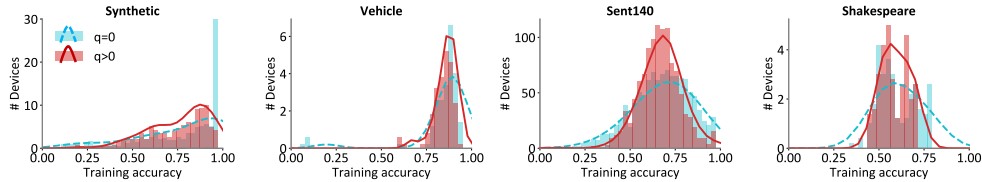

Figure 6: $q$-FFL ($q > 0$) results in more centered (i.e., fair) *training* accuracy distributions across devices without sacrificing the average accuracy.

Table 6: $q$-FFL results in more fair training accuracy distributions in terms of all uniformity measurements—(a) the accuracy variance, (b) the cosine similarity (i.e., angle) between the accuracy distribution and the all-ones vector $\mathbf{1}$, and (c) the KL divergence between the normalized accuracy $a$ and uniform distribution $u$.

| Dataset | Objective | Average (%) | Worst 10% (%) | Best 10% (%) | Variance | Angle (°) | KL($a\|\|u$) |
|---|---|---|---|---|---|---|---|
| Synthetic | $q = 0$ | $81.7 \pm .3$ | $23.6 \pm 1.1$ | $100.0 \pm .0$ | $597 \pm 10$ | $17.5 \pm .3$ | $.061 \pm .002$ |
| | $q = 1$ | $78.9 \pm .2$ | $\mathbf{41.8} \pm 1.0$ | $96.8 \pm .5$ | $\mathbf{292} \pm 11$ | $\mathbf{12.5} \pm .2$ | $\mathbf{.027} \pm .001$ |
| Vehicle | $q = 0$ | $87.5 \pm .2$ | $49.5 \pm 10.2$ | $\mathbf{94.9} \pm .7$ | $237 \pm 97$ | $10.2 \pm 2.4$ | $.025 \pm .011$ |
| | $q = 5$ | $87.8 \pm .5$ | $\mathbf{71.3} \pm 2.2$ | $93.1 \pm 1.4$ | $\mathbf{37} \pm 12$ | $\mathbf{4.0} \pm .7$ | $\mathbf{.003} \pm .001$ |
| Sent140 | $q = 0$ | $69.8 \pm .8$ | $36.9 \pm 3.1$ | $\mathbf{94.4} \pm 1.1$ | $278 \pm 44$ | $13.6 \pm 1.1$ | $.032 \pm .006$ |
| | $q = 1$ | $68.2 \pm .6$ | $\mathbf{46.0} \pm .3$ | $88.8 \pm .8$ | $\mathbf{143} \pm 4$ | $\mathbf{10.0} \pm .1$ | $\mathbf{.017} \pm .000$ |
| Shakespeare | $q = 0$ | $72.7 \pm .8$ | $46.4 \pm 1.4$ | $\mathbf{79.7} \pm .9$ | $116 \pm 8$ | $9.9 \pm .3$ | $.015 \pm .001$ |
| | $q = .001$ | $66.7 \pm 1.2$ | $\mathbf{48.0} \pm .4$ | $71.2 \pm 1.9$ | $\mathbf{56} \pm 9$ | $\mathbf{7.1} \pm .5$ | $\mathbf{.008} \pm .001$ |

**Average testing accuracy with respect to devices.** In Section 4.2, we show that $q$-FFL leads to more fair accuracy distributions while maintaining approximately the same testing accuracies. Note that we report average testing accuracy with respect to *all data points* in Table 1. However, we observe similar results on average accuracy with respect to *all devices* between $q = 0$ and $q > 0$ objectives, as shown in Table 7. This indicates that $q$-FFL can reduce the variance of the accuracy distribution without sacrificing the average accuracy over devices or over data points.

Table 7: Average testing accuracy under $q$-FFL objectives. We show that the resulting solutions of $q = 0$ and $q > 0$ objectives have approximately the same average accuracies both with respect to all data points and with respect to all devices.

| Dataset | Objective | Accuracy w.r.t. Data Points (%) | Accuracy w.r.t. Devices (%) |
|---|---|---|---|
| Synthetic | $q = 0$ | $80.8 \pm .9$ | $77.3 \pm .6$ |
| | $q = 1$ | $79.0 \pm 1.2$ | $76.3 \pm 1.7$ |
| Vehicle | $q = 0$ | $87.3 \pm .5$ | $85.6 \pm .4$ |
| | $q = 5$ | $87.7 \pm .7$ | $86.5 \pm .7$ |
| Sent140 | $q = 0$ | $65.1 \pm 4.8$ | $64.6 \pm 4.5$ |
| | $q = 1$ | $66.5 \pm .2$ | $66.2 \pm .2$ |
| Shakespeare | $q = 0$ | $51.1 \pm .3$ | $61.4 \pm 2.7$ |
| | $q = .001$ | $52.1 \pm .3$ | $60.0 \pm .5$ |

**Comparison with uniform sampling.** In Figure 7 and Table 8, we show that in terms of training accuracies, the uniform sampling heuristic may outperform $q$-FFL (as opposed to the testing accuracy results in Section 4). We suspect that this is because the uniform sampling baseline is a static method and is likely to overfit to those devices with few samples. In additional to Figure 3 in Section 4.3, we also report the average testing accuracy with respect to data points, best 10%, worst 10% accuracies, and the variance (along with two other uniformity measures) in Table 9.

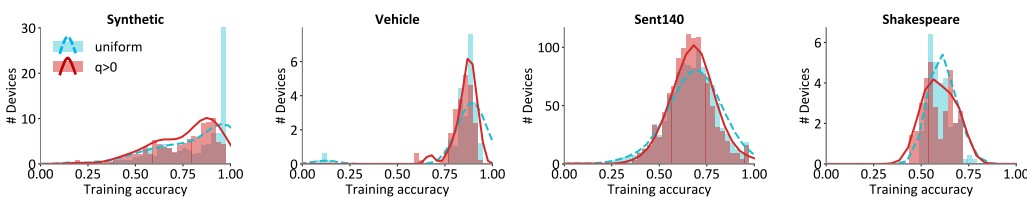

Figure 7: $q$-FFL ($q > 0$) compared with uniform sampling in training accuracy. We see that on some datasets uniform sampling has higher (and more fair) training accuracies due to the fact that it is overfitting to devices with few samples.

Table 8: More statistics comparing the uniform sampling objective with $q$-FFL in terms of training accuracies. We observe that uniform sampling could result in more fair *training* accuracy distributions with smaller variance in some cases.

| Dataset | Objective | Average (%) | Worst 10% (%) | Best 10% (%) | Variance | Angle (°) | KL($a\|u$) |
|---|---|---|---|---|---|---|---|
| Synthetic | uniform | $83.5 \pm .2$ | $\mathbf{42.6} \pm 1.4$ | $\mathbf{100.0} \pm .0$ | $366 \pm 17$ | $13.4 \pm .3$ | $.031 \pm .002$ |
| | $q = 1$ | $78.9 \pm .2$ | $41.8 \pm 1.0$ | $96.8 \pm .5$ | $\mathbf{292} \pm 11$ | $\mathbf{12.5} \pm .2$ | $\mathbf{.027} \pm .001$ |
| Vehicle | uniform | $87.3 \pm .3$ | $46.6 \pm .8$ | $\mathbf{94.8} \pm .5$ | $261 \pm 10$ | $10.7 \pm .2$ | $.027 \pm .001$ |
| | $q = 5$ | $87.8 \pm .5$ | $\mathbf{71.3} \pm 2.2$ | $93.1 \pm 1.4$ | $\mathbf{37} \pm 12$ | $\mathbf{4.0} \pm .7$ | $\mathbf{.003} \pm .001$ |
| Sent140 | uniform | $69.1 \pm .5$ | $42.2 \pm 1.1$ | $\mathbf{91.0} \pm 1.3$ | $188 \pm 19$ | $11.3 \pm .5$ | $.022 \pm .002$ |
| | $q = 1$ | $68.2 \pm .6$ | $\mathbf{46.0} \pm .3$ | $88.8 \pm .8$ | $\mathbf{143} \pm 4$ | $\mathbf{10.0} \pm .1$ | $\mathbf{.017} \pm .000$ |
| Shakespeare | uniform | $57.7 \pm 1.5$ | $\mathbf{54.1} \pm 1.7$ | $\mathbf{72.4} \pm 3.2$ | $\mathbf{32} \pm 7$ | $\mathbf{5.2} \pm .5$ | $\mathbf{.004} \pm .001$ |
| | $q = .001$ | $66.7 \pm 1.2$ | $48.0 \pm .4$ | $71.2 \pm 1.9$ | $56 \pm 9$ | $7.1 \pm .5$ | $.008 \pm .001$ |

Table 9: More statistics showing more fair solutions induced by $q$-FFL compared with the uniform sampling baseline in terms of *test* accuracies. Again, we observe that under $q$-FFL, the testing accuracy of the worst 10% devices tends to increase compared with uniform sampling, and the variance of the final testing accuracies is smaller. Similarly, $q$-FFL is also more fair than uniform sampling in terms of other uniformity metrics.

| Dataset | Objective | Average (%) | Worst 10% (%) | Best 10% (%) | Variance | Angle (°) | KL($a\|u$) |
|---|---|---|---|---|---|---|---|
| Synthetic | uniform | $82.2 \pm 1.1$ | $30.0 \pm .4$ | $100.0 \pm .0$ | $525 \pm 47$ | $\mathbf{15.6} \pm .8$ | $\mathbf{.048} \pm .007$ |
| | $q = 1$ | $79.0 \pm 1.2$ | $\mathbf{31.1} \pm 1.8$ | $100.0 \pm 0.0$ | $\mathbf{472} \pm 14$ | $16.0 \pm .5$ | $.049 \pm .003$ |
| Vehicle | uniform | $86.8 \pm .3$ | $45.4 \pm .3$ | $\mathbf{95.4} \pm .7$ | $267 \pm 7$ | $10.8 \pm .1$ | $.028 \pm .001$ |
| | $q = 5$ | $87.7 \pm 0.7$ | $\mathbf{69.9} \pm .6$ | $94.0 \pm .9$ | $\mathbf{48} \pm 5$ | $\mathbf{4.6} \pm .2$ | $\mathbf{.003} \pm .000$ |
| Sent140 | uniform | $66.6 \pm 2.6$ | $21.1 \pm 1.9$ | $100.0 \pm 0.0$ | $560 \pm 19$ | $19.8 \pm .7$ | $.076 \pm .006$ |
| | $q = 1$ | $66.5 \pm .2$ | $\mathbf{23.0} \pm 1.4$ | $100.0 \pm 0.0$ | $\mathbf{509} \pm 30$ | $\mathbf{18.8} \pm .5$ | $\mathbf{.067} \pm .006$ |
| Shakespeare | uniform | $50.9 \pm .4$ | $41.0 \pm 3.7$ | $\mathbf{70.6} \pm 5.4$ | $71 \pm 38$ | $9.1 \pm 2.8$ | $.012 \pm .006$ |
| | $q = .001$ | $52.1 \pm .3$ | $\mathbf{42.1} \pm 2.1$ | $69.0 \pm 4.4$ | $\mathbf{54} \pm 27$ | $\mathbf{7.9} \pm 2.3$ | $\mathbf{.009} \pm .05$ |

## E.2  ADDITIONAL EXPERIMENTS

**Effects of data heterogeneity and the number of devices on unfairness.**  To study how data heterogeneity and the total number of devices affect unfairness in a more direct way, we investigate into a set of synthetic datasets where we can quantify the degree of heterogeneity. The results are shown in Table 10 below. We generate three synthetic datasets following the process described in Appendix D.1, but with different parameters to control heterogeneity. In particular, we generate an IID data— Synthetic (IID) by setting the same $W$ and $b$ on all devices and setting the samples $x_k \sim \mathcal{N}(0, 1)$ for any device $k$. We instantiate two non-identically distributed datasets (Synthetic (1, 1) and Synthetic (2, 2)) from Synthetic $(\alpha, \beta)$ where $u_k \sim \mathcal{N}(0, \alpha)$ and $B_k \sim \mathcal{N}(0, \beta)$. Recall that $\alpha, \beta$ allows to precisely manipulate the degree of heterogeneity with larger $\alpha, \beta$ values indicating more statistical heterogeneity. Therefore, from top to bottom in Table 10, data are more heterogeneous. For each dataset, we further create two variants with different number of participating devices. We see that as data become more heterogeneous and as the number of devices in the network increases, the accuracy distribution tends to be less uniform.

Table 10: Effects of data heterogeneity and the number of devices on unfairness. For a fixed number of devices, as data heterogeneity increases from top to bottom, the accuracy distributions become less uniform (with larger variance) for both $q = 0$ and $q > 0$. Within each dataset, the decreasing number of devices results in a more uniform accuracy distribution. In all scenarios (except on IID data), setting $q > 0$ helps to encourage more fair solutions.

| Dataset | | Objective | Average | Worst 10% | Best 10% | Variance |
|---|---|---|---|---|---|---|
| Synthetic (IID) | 100 devices | $q = 0$ | $89.2 \pm .6$ | $70.9 \pm 3$ | $100.0 \pm 0$ | $85 \pm 15$ |
| | | $q = 1$ | $89.0 \pm .5$ | $70.3 \pm 3$ | $100.0 \pm 0$ | $88 \pm 19$ |
| | 50 devices | $q = 0$ | $87.1 \pm 1.5$ | $66.5 \pm 3$ | $100.0 \pm 0$ | $107 \pm 14$ |
| | | $q = 1$ | $86.8 \pm 0.8$ | $66.5 \pm 2$ | $100.0 \pm 0$ | $109 \pm 13$ |
| Synthetic (1, 1) | 100 devices | $q = 0$ | $83.0 \pm .9$ | $36.8 \pm 2$ | $100.0 \pm 0$ | $452 \pm 22$ |
| | | $q = 1$ | $82.7 \pm 1.3$ | $\mathbf{43.5} \pm 5$ | $100.0 \pm 0$ | $\mathbf{362} \pm 58$ |
| | 50 devices | $q = 0$ | $84.5 \pm .3$ | $43.3 \pm 2$ | $100.0 \pm 0$ | $370 \pm 37$ |
| | | $q = 1$ | $85.1 \pm .8$ | $\mathbf{47.3} \pm 3$ | $100.0 \pm 0$ | $\mathbf{317} \pm 41$ |
| Synthetic (2, 2) | 100 devices | $q = 0$ | $82.6 \pm 1.1$ | $25.5 \pm 8$ | $100.0 \pm 0$ | $618 \pm 117$ |
| | | $q = 1$ | $82.2 \pm 0.7$ | $\mathbf{31.9} \pm 6$ | $100.0 \pm 0$ | $\mathbf{484} \pm 79$ |
| | 50 devices | $q = 0$ | $85.9 \pm 1.0$ | $36.8 \pm 7$ | $100.0 \pm 0$ | $421 \pm 85$ |
| | | $q = 1$ | $85.9 \pm 1.4$ | $\mathbf{39.1} \pm 6$ | $100.0 \pm 0$ | $\mathbf{396} \pm 76$ |

**A family of $q$'s results in variable levels of fairness.**  In Table 11, we show the accuracy distribution statistics of using a family of $q$'s on synthetic data. Our objective and methods are not sensitive to any particular $q$ since all $q > 0$ values can lead to more fair solutions compared with $q = 0$. In our experiments in Section 4, we report the results using the $q$ values selected following the protocol described in Appendix D.2.3.

Table 11: Test accuracy statistics of using a family of $q$'s on synthetic data. We show results with $q$'s selected from our candidate set $\{0.001, 0.01, 0.1, 1, 2, 5, 10, 15\}$. $q$-FFL allows for a more flexible trade-off between fairness and accuracy. A larger $q$ results in more fairness (smaller variance), but potentially lower accuracy. Similarly, a larger $q$ imposes more uniformity in terms of other metrics—(a) the cosine similarity/angle between the accuracy distribution and the all-ones vector $\mathbf{1}$, and (b) the KL divergence between the normalized accuracy $a$ and a uniform distribution $u$.

| Dataset | Objective | Average (%) | Worst 10% (%) | Best 10% (%) | Variance | Angle (°) | KL($a\|u$) |
|---|---|---|---|---|---|---|---|
| Synthetic | $q=0$ | $80.8 \pm .9$ | $18.8 \pm 5.0$ | $100.0 \pm 0.0$ | $724 \pm 72$ | $19.5 \pm 1.1$ | $.083 \pm .013$ |
| | $q=0.1$ | $81.1 \pm 0.8$ | $22.1 \pm .8$ | $100.0 \pm 0.0$ | $666 \pm 56$ | $18.4 \pm .8$ | $.070 \pm .009$ |
| | $q=1$ | $79.0 \pm 1.2$ | $31.1 \pm 1.8$ | $100.0 \pm 0.0$ | $472 \pm 14$ | $16.0 \pm .5$ | $.049 \pm .003$ |
| | $q=2$ | $74.7 \pm 1.3$ | $32.2 \pm 2.1$ | $99.9 \pm .2$ | $410 \pm 23$ | $15.6 \pm 0.7$ | $.044 \pm .005$ |
| | $q=5$ | $67.2 \pm 0.9$ | $30.0 \pm 4.8$ | $94.3 \pm 1.4$ | $369 \pm 51$ | $16.3 \pm 1.2$ | $.048 \pm .010$ |

**Device-specific $q$.** In these experiments, we explore a device-specific strategy for selecting $q$ in $q$-FFL. We solve $q$-FFL with $q \in \{0, 0.001, 0.01, 0.1, 1, 2, 5, 10\}$ in parallel. After training, each device selects the best resulting model based on the validation data and tests the performance of the model using the testing set. We report the results in terms of testing accuracy in Table 12. Interestingly, using this device-specific strategy the average accuracy in fact increases while the variance of accuracies is reduced, in comparison with $q = 0$. We note that this strategy does induce more local computation and additional communication load at each round. However, it does not increase the number of communication rounds if run in parallel.

Table 12: Effects of running $q$-FFL with several $q$'s in parallel. We train multiple global models (corresponding to different $q$'s) independently in the network. After the training finishes, each device picks up a best, device-specific model based on the performance (accuracy) on the validation data. While this adds additional local computation and more communication load per round, the device-specific strategy has the added benefit of increasing the accuracies of devices with the worst 10% accuracies and devices with the best 10% accuracies simultaneously. This strategy is built upon the proposed primitive Algorithm 2, and in practice, people can develop other heuristics to improve the performance (similar to what we explore here), based on the method of adaptively averaging model updates proposed in Algorithm 2.

| Dataset | Objective | Average (%) | Worst 10% (%) | Best 10% (%) | Variance | Angle (°) | KL($a\|u$) |
|---|---|---|---|---|---|---|---|
| Vehicle | $q=0$ | $87.3 \pm .5$ | $43.0 \pm 1.0$ | $95.7 \pm 1.0$ | $291 \pm 18$ | $11.3 \pm .3$ | $.031 \pm .003$ |
| | $q=5$ | $87.7 \pm .7$ | $69.9 \pm .6$ | $94.0 \pm .9$ | $48 \pm 5$ | $4.6 \pm .2$ | $.003 \pm .000$ |
| | multiple $q$ | $88.5 \pm .3$ | $70.0 \pm 2.0$ | $95.8 \pm .6$ | $52 \pm 7$ | $4.7 \pm .3$ | $.004 \pm .000$ |
| Shakespeare | $q=0$ | $51.1 \pm .3$ | $39.7 \pm 2.8$ | $72.9 \pm 6.7$ | $82 \pm 41$ | $9.8 \pm 2.7$ | $.014 \pm .006$ |
| | $q=.001$ | $52.1 \pm .3$ | $42.1 \pm 2.1$ | $69.0 \pm 4.4$ | $54 \pm 27$ | $7.9 \pm 2.3$ | $.009 \pm .05$ |
| | multiple $q$ | $52.0 \pm 1.5$ | $41.0 \pm 4.3$ | $72.0 \pm 4.8$ | $72 \pm 32$ | $10.1 \pm .7$ | $.017 \pm .000$ |

**Convergence speed of $q$-FFL.** Since q−FFL $(q > 0)$ is more difficult to optimize, a natural question one might ask is: will the $q$-FFL $q > 0$ objectives slow the convergence compared with FedAvg? We empirically investigate this on the four datasets. We use $q$-FedAvg to solve $q$-FFL, and compare it with FedAvg (i.e., solving $q$-FFL with $q = 0$). As demonstrated in Figure 8, the $q$ values that result in more fair solutions also do not significantly slow down convergence.

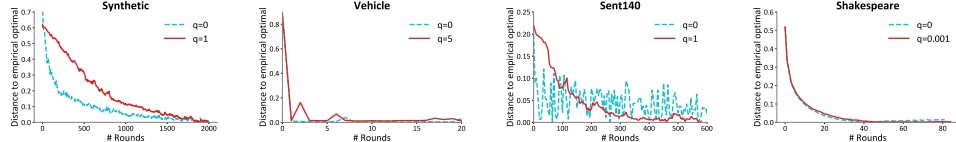

Figure 8: The convergence speed of $q$-FFL compared with FedAvg. We plot the distance to the highest accuracy achieved versus communication rounds. Although $q$-FFL with $q>0$ is a more difficult optimization problem, for the $q$ values we choose that could lead to more fair results, the convergence speed is comparable to that of $q = 0$.

**Efficiency of $q$-FFL compared with AFL.** One added benefit of $q$-FFL is that it leads to faster convergence than AFL—even when we use *non-local-updating* methods for both objectives. In Figure 9, we show with respect to the final testing accuracy for the single worst device (i.e., the objective that AFL is trying to optimize), $q$-FFL converges faster than AFL. As the number of devices increases (from Fashion MNIST to Vehicle), the performance gap between AFL and $q$-FFL becomes larger because AFL introduces larger variance.

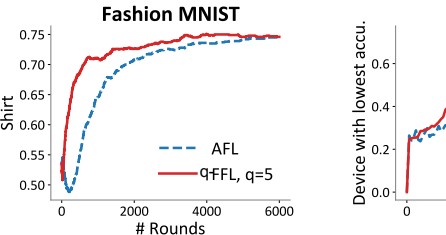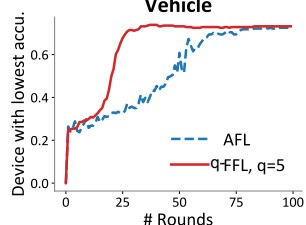

Figure 9: $q$-FFL is more efficient than AFL. With the worst device achieving the same final testing accuracy, $q$-FFL converges faster than AFL. For Vehicle (with 23 devices) as opposed to Fashion MNIST (with 3 devices), we see that the performance gap is larger. We run full gradient descent at each round for both methods.

**Efficiency of $q$-FedAvg under different data heterogeneity.** As mentioned in Appendix E.1, one potential cause for the slower convergence of $q$-FedAvg on the synthetic dataset may be that local updating schemes could hurt convergence when local data distributions are highly heterogeneous. Although it has been shown that applying updates locally results in significantly faster convergence in terms of communication rounds (McMahan et al., 2017; Smith et al., 2018), which is consistent with our observation on most datasets, we note that when data is highly heterogeneous, local updating may hurt convergence. We validate this by creating an IID synthetic dataset (Synthetic-IID) where local data on each device follow the same global distribution. We call the synthetic dataset used in Section 4 Synthetic-Non-IID. We also create a hybrid dataset (Synthetic-Hybrid) where half of the total devices are assigned IID data from the same distribution, and half of the total devices are assigned data from different distributions. We observe that if data is perfectly IID, $q$-FedAvg is more efficient than $q$-FedSGD. As data become more heterogeneous, $q$-FedAvg converges more slowly than $q$-FedSGD in terms of communication rounds. For all three synthetic datasets, we repeat the process of tuning a best constant step-size for FedSGD and observe similar results as before — our dynamic solver $q$-FedSGD behaves similarly (or even outperforms) a best hand-tuned FedSGD.

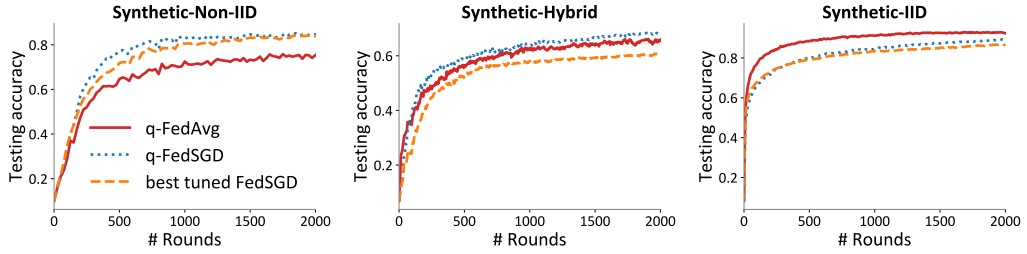

Figure 10: Convergence of $q$-FedAvg compared with $q$-FedSGD under different data heterogeneity. When data distributions are heterogeneous, it is possible that $q$-FedAvg converges more slowly than $q$-FedSGD. Again, the proposed dynamic solver $q$-FedSGD performs similarly (or better) than a best tuned fixed-step-size FedSGD.

