# OpenReview forum: "Fair Resource Allocation in Federated Learning"
_ICLR.cc/2020/Conference — Accept (Poster)_

### Official Review · AnonReviewer1 · 2019-10-22
**Official Blind Review #1**

**Rating:** 6

**Review:**

In this paper, the authors propose a new optimization objective for fair resource allocation. Furthermore, a new algorithm, q-FedAvg, based on the vanilla federated learning, is proposed to solve the new optimization in massive and heterogeneous networks. The paper is well written. Theoretical analysis is also provided to support the effectiveness of the proposed methods. The experiments show good performance.
In overall, I think this paper solves an important problem in federated learning, and I vote for acceptance.
However, since my knowledge in fairness is very limitted, I think my review is an educated guess. If the other reviews vote for rejection, I will not champion this paper.

I have question to the authors:

Is the proposed algorithm robust to the estimation of the Lipschitz constant? In my opinion, the proposed algorithm highly relies on $L_q(w)$. Thus, the estimation of L will be very essential. It will be better if the authors can show some results where different estimations of L  is used, and compare these results to show the sensitivity to the estimation of L.

**Experience Assessment:**

I do not know much about this area.

**Review Assessment: Checking Correctness Of Derivations And Theory:**

I did not assess the derivations or theory.

**Review Assessment: Checking Correctness Of Experiments:**

I assessed the sensibility of the experiments.

**Review Assessment: Thoroughness In Paper Reading:**

I made a quick assessment of this paper.

---

> ### Author Response · Authors · 2019-11-13
> **Response to Reviewer #1**
>
> We thank the reviewer for the positive evaluation of our work.
>
> [Sensitivity of L] The reviewer is correct that it is important to derive a good estimate of L; similar to other first-order methods (including FedAvg), our algorithms (q-FedAvg and q-FedSGD) are sensitive to the choice of local learning rate, and are therefore sensitive to the estimate of the local Lipschitz constant L. However, we note that when q=0, our method does not incur any additional hyperparameter tuning cost beyond what is necessary for other first-order methods (such as FedAvg). The benefit of our approach is that when q>0, we suggest using the Lipschitz constant estimated at q=0 to easily derive an appropriate learning rate for q>0. This removes the need for tuning the learning rate at q>0. To test the robustness of this approach, we explore the efficacy of this heuristic directly in Figure 3. In particular, we compare q-FedSGD (with an estimated L) to FedSGD using the best-tuned learning rate for the same q. As is evident from Figure 3, this simple heuristic does not result in any performance degradation compared to the best tuned step-size, which helps to motivate the use of our approach.

---

### Official Review · AnonReviewer2 · 2019-10-26
**Official Blind Review #2**

**Rating:** 3

**Review:**

[Summary]
The authors propose a protocol to encourage a more fair distribution of the performance across devices in a federated setting. In contrast with previous work, which protects a specific attribute, this paper aims to achieve the uniformity of the accuracy distribution.

[Key Comments]
The paper is well-organized and clearly written. The claims are well-supported by theoretical analysis and experimental results. However, my main concern is that the paper offers an incremental improvement over the early work FedAvg (McMahan et al., 2017). It would be helpful for the authors to summarize their contributions if space permits.

[Details]
[Pro 1] This paper provides insights into fairness (a more uniform accuracy distribution) in federated learning, which appears to be well-motivated.

[Pro 2] This paper provides an instructive method to estimate the upper-bound of the Lipschitz constants for ??? the local objective function (the objective function with clients' data) ???. It is an interesting idea to choose dynamic step-size depending on the global Lipschitz constants and fairness parameter q.

[Pro 3] The evaluation fully considers various uniformity metrics, sampling strategies, and the chosen of q.

[Con 1] I am confused about the difference between the proposed method and Newton's method. It would be helpful for the authors to clarify the limitation of the objective function (for example, the objective function should be second-order derivable).

[Con 2] The authors note that "It is not straightforward to simply apply FedAvg to problem (2) when q>0, as the F_{k}^{q+1} term prevents the use of local SGD." I found it difficult for me to follow this argument. Is it relevant to the parameter q? Given the communication-efficiency improvement in Section 3.3, few explanations are provided for the main improvement over previous work. Is it because of the local updating? Otherwise, more details about the convergence rate will strengthen the submission.

**Experience Assessment:**

I have read many papers in this area.

**Review Assessment: Checking Correctness Of Derivations And Theory:**

I carefully checked the derivations and theory.

**Review Assessment: Checking Correctness Of Experiments:**

I carefully checked the experiments.

**Review Assessment: Thoroughness In Paper Reading:**

I read the paper thoroughly.

---

> ### Author Response · Authors · 2019-11-13
> **Response to Reviewer #2**
>
> We thank the reviewer for their careful review of the paper. As per the reviewer’s key comment, please see the [Contributions] section in our response to all reviewers.
>
> [Differences with Newton’s method] Unlike Newton’s method, which uses the second-order derivative information, our algorithm is a first-order method which only relies on the gradient information. It is common to use first-order methods in large-scale optimization (particularly for applications in federated learning) as second-order methods can be costly. Please let us know if this does not answer your question.
>
> [q-FedAvg] Thanks for these questions regarding our method; we answer them here, and have also updated Section 3.3 in our revision to better explain our approach.
> (1) FedAvg is applicable to objectives where each local loss function is an empirical average over the loss on all local data points. Unfortunately, for our q-FFL objective, when q>0, $F_k^{q+1}$ is not a simple empirical average of the loss of local samples due to the (q+1) exponent. We therefore propose to generalize FedAvg for non-zero q by using a more sophisticated dynamic weighted average scheme, which is explained in Alg. 1 (q-FedAvg). We note that at q=0, the dynamic weighting in q-FedAvg simplifies to simple averaging, which recovers FedAvg as a special case.
> (2) The reviewer is correct that the communication improvements of q-FedAvg over q-FedSGD are due to the local updating scheme; we discuss this in the last paragraph of Section 3.3. The benefits of local updating approaches such as FedAvg have been shown in simpler settings, which is what motivated us to adapt this approach to our q-FFL objective.

---

### Official Review · AnonReviewer3 · 2019-10-28
**Official Blind Review #3**

**Rating:** 3

**Review:**

The problem of fairness in federated learning (FL) is important given the popularity of the topic and its immediate impact on the society. Vanilla FL approaches may be subject to poor performance for clients whose data is under-represented across all participants. This paper proposes a new algorithm for federated learning to reduce variance in performance across clients. The inspiration for the algorithm comes from the problem of uniform  resource allocation in wireless networks.

While the problem and the motivation for the algorithm are interesting on the high level, I think this paper does not deliver the key ideas in sufficient detail and clarity.

On the algorithms side, I am still unclear on how the Lipschitz constant L is estimated on the first run with q=0. Are the results for q=0 in the experiments reported for this run or is it repeated with the learned L? Further, this procedure suggests that the number of communication rounds is at least doubled for the end-to-end training. Tuning q, which seems to be necessary, may require even more communication rounds.

While there are a lot of experiments in the paper (across main text and supplementary), none seem to be carried out sufficiently well. Understanding the complete experimental setup for at least one of them is also quite hard due to numerous supplementary references throughout the experiments section. I would recommend to focus on fewer experiments, but present more thorough results. Below are some suggestions.

The importance of resource allocation in FL appears to me to be directly related to the key FL aspects such as degree of data heterogeneity and number of clients. This submission is lacking experiments comparing FedAvg to the proposed method under these settings (which can be simulated using available datasets). To argue in favor of the proposed approach it is important to demonstrate failure modes of the existing algorithms under some realistic scenarios and present a solution using new algorithm.

Accuracies in Fashion MNIST and Shakespeare experiments seem quite poor suggesting some problems with the setup. FedAvg paper reports 54% on Shakespeare, whereas this paper reports 52%. It also appears that the number of considered "devices" on Shakespeare is significantly smaller than in the FedAvg paper (31 vs 1146) - what is the reason for this?
On Fashion MNIST, AFL paper reports 80%+ accuracy while achieving 90%+ on the combined dataset seems relative easy based on the results mentioned on the Github repository of the dataset. This paper reports 78% for the proposed method and AFL. Why is there a discrepancy with AFL paper and what is the performance of FedAvg on this dataset (assuming some suitable CNN architecture)? Is there a reason to believe that this dataset is much harder for federated learning than MNIST, where FedAvg roughly matches full data training?

This statement is ambiguous "uniform sampling is a static method and can easily overfit to devices with very few data points, whereas q-FFL has better generalization properties due to its dynamic nature." If there is a device with very few data points it is easy to overfit to it and q-FFL will essentially ignore that device since the loss on this device is very small. Why does this not lead to more severe overfitting behavior?

**Experience Assessment:**

I have published one or two papers in this area.

**Review Assessment: Checking Correctness Of Derivations And Theory:**

I assessed the sensibility of the derivations and theory.

**Review Assessment: Checking Correctness Of Experiments:**

I assessed the sensibility of the experiments.

**Review Assessment: Thoroughness In Paper Reading:**

I read the paper at least twice and used my best judgement in assessing the paper.

---

> ### Author Response · Authors · 2019-11-13
> **Response to Reviewer #3 (Part1)**
>
>
> We greatly appreciate the reviewer’s detailed review and suggestions to improve the paper.
>
> [Estimating the Lipschitz constant and tuning q] For a discussion on how to estimate the Lipschitz constants, please see our response to all reviewers; we have also clarified this in Section 3.3 of our revision. The results for q=0 are repeated with the estimated L. Note that our heuristic does not add any out of the ordinary computational cost as the learning rate (for q=0 which corresponds to FedAvg) is typically tuned via grid search. The reviewer is correct that the flexibility of having a tunable q to allow for tradeoffs between fairness and average performance comes at a cost of additional communication rounds. However, our proposed method helps to significantly reduce the number of communication rounds by obviating the need for tuning a learning rate for additional values of q>0.
>
> [Experiment setup] We include three key experiments in the main paper. Figure 1 + Table 1 together demonstrate that our objective is more ‘fair’ compared with FedAvg (confirming our theory). Figure 2 and Table 2 further show that our objective is more fair/more flexible compared with other baselines that may also lead to fairness. Finally, Figure 3 shows the efficiency of our method. The experiments in the appendix include more minor results such as how our objective performs on the training data. We have added an overview at the beginning of the appendix to make it easier to navigate. As per your suggestion, we have also included new experiments that directly explore the impact of data heterogeneity and number of clients on q-FFL.
>
> [Impact of data heterogeneity and the number of clients on q-FFL] In our existing experiments, we present failure modes by showing high degrees of performance variance when using FedAvg on a number of real-world datasets, which naturally vary in terms of heterogeneity and the number of clients. However, to more directly evaluate the effect of data heterogeneity and number of devices, we have performed new experiments on synthetic data where the heterogeneity can be quantified more precisely. Below, we show the results of the test accuracy (%) distribution averaged across 5 random test-val-train partitions of each dataset. We have also updated the paper and report the results in Table 9 in Appendix F.2. We are happy to move them to the main text if needed.
>
> Table 1. Effects of data heterogeneity and number of devices on unfairness.
> ——————————————————————————————————————
>               Dataset                     |objective| average  | worst 10%| best 10% | variance
> ——————————————————————————————————————
> Synthetic (iid)| 100 devices|  q = 0      | 89.2 (0.6)|  70.9 (3)    | 100.0 (0)  |  85 (15)
>                          |  	                 |  q = 1     | 89.0 (0.5) |  70.3 (3)    | 100.0 (0)  |  88 (19)
>                          |-------------------------------------------------------------------------------------------
>  	                 |  50 devices |  q = 0     | 87.1 (1.5) |  66.5 (3)   | 100.0 (0)  | 107 (14)
>                          |                      |  q = 1     | 86.8 (0.8) |  66.5 (2)   | 100.0 (0)  | 109 (13)
> ——————————————————————————————————————
> Synthetic (1,1)|100 devices | q = 0     | 83.0 (0.9) |  36.8 (2)   | 100.0 (0)  | 452 (22)
>                          |                      |  q = 1    | 82.7 (1.3) |  43.5 (5)   | 100.0 (0)  | 362 (58)
>                          |-------------------------------------------------------------------------------------------
>  	                 |  50 devices |  q = 0    | 84.5 (0.3) |  43.3 (2)    | 100.0 (0)  | 370 (37)
>                          |                      |  q = 1    | 85.1 (0.8) |  47.3 (3)    | 100.0 (0)  | 317 (41)
> ——————————————————————————————————————
> Synthetic (2,2)| 100 devices | q = 0   | 82.6 (1.1) |  25.5 (8)    | 100.0 (0)  | 618 (117)
>                          |                       | q = 1   | 82.2 (0.7) |  31.9 (6)    | 100.0 (0)  | 484 (79)
>                          |--------------------------------------------------------------------------------------------
>  	                 |  50 devices  | q = 0    | 85.9 (1.0)  |  36.8 (7)    | 100.0 (0)  | 421 (85)
>                          |                       | q = 1    | 85.9 (1.4)  |  39.1 (6)   | 100.0 (0)  | 396 (76)
> ——————————————————————————————————————
> The synthetic data is generated in a similar way as described in Appendix E.1. The degree of heterogeneity of the Synthetic (a, b) dataset becomes larger as a and b get larger. We can see that fixing the number of devices, as data become more heterogeneous, the accuracy variance (for both q=0 and q>0) also increases. We also investigate the effects of the number of devices using the same set of synthetic datasets, but with fewer number of devices (reduced from 100 to 50). We see that as the number of devices decreases, the accuracy distribution tends to be more uniform with a smaller variance.

---

> > ### Author Response · Authors · 2019-11-13
> > **Response to Reviewer #3 (Part2)**
> >
> >
> > [Accuracy discrepancies with previous work]
> > We note that the goal of our experiments is not to show superior accuracy on a certain benchmark, but rather to show that our q-FFL objective and our proposed algorithms provide a flexible tradeoff between performance (accuracy) and fairness on a range of datasets. The reviewer is correct that we have slight differences in accuracy relative to prior work, which we explain below.
> >
> > (1) For Shakespeare, we achieve 52% accuracy which is slightly lower than the numbers reported in FedAvg (54%). This is because we are using a randomly subsampled version of the dataset due to resource constraints (it takes, for example, 2 days to run and tune hyperparameters on this small dataset using 10 GPUs). It is important to note that having fewer clients tends to result in a more uniform accuracy distribution (as we showed in the previous experiments), which in fact makes it a more difficult baseline. However, we also provide results on a larger version with 260 devices to better match prior work (see table below). Similar to all of our results, q>0 leads to a more uniform accuracy distribution while maintaining similar average accuracy. Note that as we expect, the variance is larger for a larger number of devices due to the potentially increased heterogeneity.
> >
> > Table 2. Effects of q-FFL with q>0 on a larger Shakespeare dataset
> > —————————————————————————————————
> > Dataset             | objective  | average | worst 10% | best 10% | variance
> > —————————————————————————————————
> > Shakesepeare  |  q = 0        |     52.3    |    31.0         |   67.6        |     108
> >                            |  q = .001   |     52.2    |    34.5         |   67.1        |      76
> > —————————————————————————————————
> >
> > (2) For Fashion MNIST, we follow the exact setting described in the AFL paper where only 3 classes are subsampled for evaluation using a logistic regression model. We have tried our best to accurately replicate their setup including using the same optimizer, but as the code is not open-sourced, we do not have access to AFL’s exact implementation (hyperparameters, data preprocessing, etc), and are therefore uncertain as to why this discrepancy exists. We note that we have open-sourced our own code to avoid such issues with our work.
> >
> > [q-FFL is less likely to overfit] Thanks for this question; we have updated the paper to explain this statement in more detail. Unlike static weighting in uniform sampling, q-FFL dynamically gives less importance to a device as soon as its loss reduces. Thus, it is less likely to overfit to one device at the expense of not reducing the loss for other devices.

---

### Author Response · Authors · 2019-11-13
**Response to all reviewers**


We thank all reviewers for their time and helpful comments. We first address shared concerns and then respond to specific comments below.  We have updated the paper (with edits highlighted in red). In response to the reviewers, in our revision we have further clarified our proposed methods and included an additional experiment to directly explore the effects of heterogeneity and network size.

[Contributions]
In this paper, we propose q-FFL, a novel objective that encourages a fairer (more uniform) performance distribution in federated learning. This is the first work we are aware of to explore such an objective in the context of distributed machine learning. q-FFL, parameterized by q, enables a flexible fairness/accuracy tradeoff and generalizes prior work of FedAvg (q=0) and AFL ($q \to \infty$). We theoretically prove that q-FFL improves uniformity, and develop scalable methods to solve q-FFL efficiently in federated networks by dynamically combining the model updates on the central server. Our empirical evaluation on a set of real-world datasets in realistic federated scenarios demonstrates the fairness and flexibility of our objective and the efficiency of our methods.

[Methods] We clarify two aspects of our proposed methods.
- [q-FedSGD and q-FedAvg] We propose two methods to solve our objective: q-FedSGD and q-FedAvg. q-FedAvg is a communication-efficient variant of q-FedSGD that allows for local updating by replacing the gradients with model updates. This modification is analogous to the difference between distributed SGD and FedAvg, and is a common strategy in large-scale optimization to reduce communication and improve convergence speed [1].
- [Estimating the Lipschitz constant and tuning q] We crudely estimate the local Lipschitz constant on q=0 as the inverse of the best tuned learning rate using grid search, which is a heuristic commonly used in practice for many first-order optimization methods [2, 3]. We then propose to use this estimated Lipschitz value to bypass the step-size tuning for any q>0. For a practitioner who might have to find the best fairness/accuracy trade-off from a number of objectives with varying values of q, such a shortcut can save time as there would be no need to tune the learning rate for different values of q>0. We empirically verify that this heuristic works well in practice in Section 4 (please also see our response to Reviewer #1 for additional details).

[1] S. Stitch. Local SGD converges fast and communicates little. ICLR, 2019.
[2] S. Ghadimi and G. Lan. Stochastic first-and zeroth-order methods for nonconvex stochastic programming. SIAM Journal on Optimization, 2013.
[3] Y. Nesterov. Introductory lectures on convex optimization: A basic course. Springer Science & Business Media, 2013.

---

### Decision · Program_Chairs · 2019-12-19

**Decision:**

Accept (Poster)

**Comment:**

This manuscript proposes and analyzes a federated learning procedure with more uniform performance across devices, motivated as resulting in a fairer performance distribution. The resulting algorithm is tunable in terms of the fairness-performance tradeoff and is evaluated on a variety of datasets.

The reviewers and AC agree that the problem studied is timely and interesting, as there is limited work on fairness in federated learning. However, this manuscript also received quite divergent reviews, resulting from differences in opinion about the novelty and clarity of the conceptual and empirical results. In reviews and discussion, the reviewers noted insufficient justification of the approach and results, particularly in terms of broad empirical evaluation, and sensitivity of the results to misestimation of various constants. In the opinion of the AC, while the paper can be much improved, it seems to be technically correct, and the results are of sufficiently broad interest to consider publication.